# Deliberate Evolution:
# Agentic Reasoning for Sample-Efficient Symbolic Regression with LLMs

**Xinyu Pang**[1*] **Zhanke Zhou**[2*] **Xuan Li**[2] **Fangrui Lv**[1] **Shanshan Wei**[3]
**Sen Cui**[1] **Bo Han**[2] **Changshui Zhang**[1]

## Abstract

Symbolic regression (SR) discovers compact mathematical expressions from data, yet recent LLM-based evolutionary methods remain sample-inefficient because they rely mainly on scalar feedback such as MSE. We identify a core limitation: existing methods conflate *candidate proposal* with *search guidance*, requiring the LLM to infer how to evolve an expression, diagnose its errors, and reuse past experience from a single score. To address this, we propose **Deliberate Evolution** (DE), an agentic framework that decouples symbolic generation from search control. DE guides LLM proposals with adaptive operators for search direction, analytical tools for structural diagnosis, and reflective memory for trajectory-level experience. Experiments on LLM-SRBench show that DE consistently outperforms representative LLM-based SR baselines across diverse scientific domains while using only $40\%$ of the standard sample budget. Code is available at https://github.com/Xinyu-Pang/Deliberate-Evolution.

## 1. Introduction

Symbolic regression (SR) seeks compact mathematical expressions that explain observed data, making it a key tool for interpretable scientific discovery (Koza, 1994; Schmidt & Lipson, 2009). Recent LLM-based methods formulate SR as an evolutionary loop (Fig. 1): an LLM proposes symbolic candidates, numerical optimizers fit their constants, and external evaluation scores the completed expressions (Shojaee

*Equal contribution [1] Beijing National Research Center for Information Science and Technology (BNRist), Department of Automation, Tsinghua University, Beijing, P.R. China [2]TMLR Group, Department of Computer Science, Hong Kong Baptist University [3]Lenovo Research. Correspondence to: Changshui Zhang <zcs@tsinghua.edu.cn>.

*Proceedings of the $43^{rd}$ International Conference on Machine Learning*, Seoul, South Korea. PMLR 306, 2026. Copyright 2026 by the author(s).

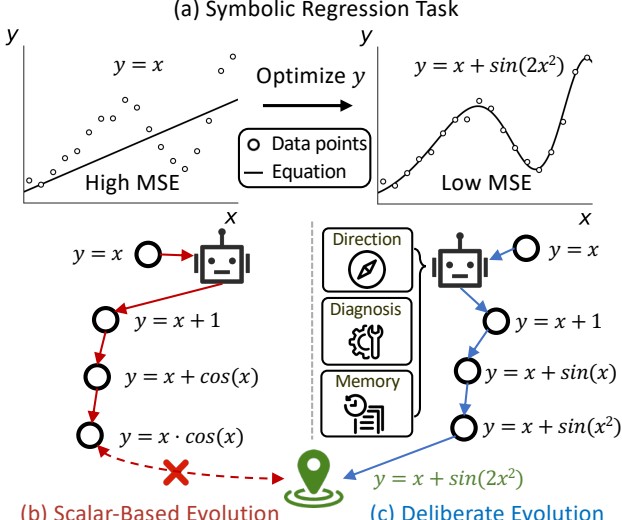

*Figure 1.* **Deliberate Evolution for symbolic regression.** (a) Symbolic regression seeks an interpretable equation that explains observed data. (b) Existing scalar-based evolution relies primarily on MSE feedback, yielding weakly guided trial-and-error search. (c) Deliberate Evolution augments candidate proposals with direction, diagnosis, and memory, enabling structured symbolic edits and more accurate equation recovery.

et al., 2025a; Grayeli et al., 2024; Lange et al., 2025). This paradigm combines LLMs' mathematical priors with verifiable numerical feedback, yet its practical utility is limited by poor *sample efficiency*: existing LLM-based SR systems typically require $10^3$ evaluated candidates per problem.

We attribute this inefficiency to a fundamental design issue: current methods conflate *candidate proposal* with *search guidance*. Given only a parent expression and a scalar score such as MSE, the LLM is expected to infer three things at once: how the expression should be modified, why it currently fails, and what lessons from previous attempts should guide the next proposal. This scalar-driven loop lacks three crucial signals needed for efficient search:

- **Direction.** A scalar score does not indicate what type of symbolic move is appropriate. The model must guess whether to refine a promising structure, introduce a larger mutation, recombine useful subexpressions, or abandon the current region and restart.

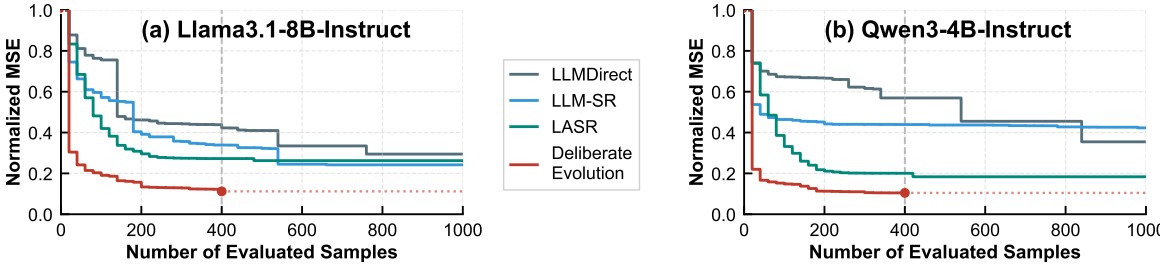

*Figure 2.* **Sample efficiency on LSR-Transform.** Normalized MSE as a function of the number of evaluated candidates using Llama3.1-8B and Qwen3-4B backbone models. Deliberate Evolution achieves lower error with substantially fewer evaluations than prior LLM-based SR methods, demonstrating superior sample efficiency.

- **Diagnosis.** MSE measures how far an expression is from the data, but not why it fails. It cannot reveal whether the error comes from missing periodicity, wrong variable interactions, invalid dimensional structure, incorrect asymptotics, or other localized structural defects.

- **Memory.** Each proposal is usually treated as a nearly independent trial. The search therefore lacks an explicit mechanism to remember which edits repeatedly fail, which symbolic motifs recur across good candidates, and which transformations have produced major improvements.

Without direction, the search may drift among unproductive edits; without diagnosis, it cannot target the structural source of error; and without memory, it repeatedly revisits failures instead of accumulating experience. Consequently, many evaluations are spent on plausible but uninformative candidates, including repetitive local variants, structurally invalid mutations, or surrogate formulas that fit observed samples without recovering the underlying law.

Our central insight is to separate *what expression to propose* from *how the search should evolve* (Fig. 1). Scalar-based evolution treats discovery as weakly guided trial and error, where candidates are proposed and judged mainly by MSE. In contrast, *deliberate evolution* makes the search trajectory explicit: directional guidance determines the symbolic move, diagnostic guidance localizes structural evidence from data and residuals, and historical guidance reuses lessons from prior attempts. This separation transforms SR from scalar-driven generation into guided scientific search.

We propose **Deliberate Evolution** (DE), an agentic framework that operationalizes this principle. In DE, the LLM acts as a flexible proposer of equation skeletons, while explicit modules steer, diagnose, and accumulate the search:

- **Adaptive operators** steer the search by explicitly choosing whether to refine, mutate, crossover, or regenerate, based on the current search state and stagnation behavior.

- **Tool-augmented proposal** diagnoses the current mismatch by analyzing data statistics, residual patterns, and dimensional consistency, turning scalar error into localized structural feedback for the LLM.

- **Reflective memory** accumulates experience by summarizing successful motifs, failed edits, and stagnated trajectories, helping future proposals reuse promising structures and avoid redundant exploration.

Together, these mechanisms decouple symbolic generation from search control, so each proposal is informed by explicit intent, localized evidence, and accumulated experience.

We evaluate Deliberate Evolution on LLM-SRBench (Shojaee et al., 2025b), covering LSR-Transform and LSR-Synth across Physics, Material Science, Chemistry, and Biology. With Llama3.1-8B-Instruct (Dubey et al., 2024) and Qwen3-4B-Instruct (Yang et al., 2025), Deliberate Evolution consistently outperforms representative LLM-based SR baselines, including LLMDirect, LLM-SR, LASR, and SGA. As shown in Fig. 2, Deliberate Evolution achieves lower error with fewer evaluated samples: using only $40\%$ of the standard LLM sampling budget, it reduces average NMSE by $55\%$ with Llama3.1-8B and by $37\%$ with Qwen3-4B. Further studies on out-of-distribution generalization, noisy observations, real-world stress-strain measurements, and ablations show that deliberate guidance improves fitting accuracy, robustness, and sample efficiency.

## 2. Preliminaries

**Problem Formulation.** Symbolic regression (SR) aims to recover an interpretable mathematical expression from observed input-output data. Given an unknown target function $f^\star : \mathbb{R}^d \to \mathbb{R}$ and a dataset $\mathcal{D} = \{(\mathbf{x}_i, y_i)\}_{i=1}^n$ with $y_i = f^\star(\mathbf{x}_i)$, SR searches over the admissible expression space $\mathcal{F}$ to find an expression that fits the observations:

$$\hat{f} = \arg\min_{f \in \mathcal{F}} \ell(f), \quad \ell(f) := \frac{1}{n} \sum_{i=1}^n \big(f(\mathbf{x}_i) - y_i\big)^2. \quad (1)$$

Here, $\mathcal{F}$ is defined by a prescribed set of variables, constants, and primitive operators, and the loss $\ell(f)$ typically corresponds to the mean squared error (MSE). The goal is not only to minimize empirical error but also to obtain an expression that is compact and generalizes to unseen inputs.

**Standard Workflow of LLM-Based Evolution.** Recent LLM-based SR methods, such as LASR (Grayeli et al., 2024), LLM-SR (Shojaee et al., 2025a), and SGA (Ma et al., 2024), commonly adopt an evolutionary optimization framework. These methods maintain a population $\mathcal{P}_t$ of candidate expressions, sometimes partitioned into multiple islands to preserve diversity and mitigate premature convergence. Each evolutionary round updates the population through selection, variation, and evaluation, detailed as follows.

In the **selection** stage, parent expressions $f_p$ are sampled from the current population $\mathcal{P}_t$ according to a fitness-dependent selection distribution $p_{\text{sel}}$:

$$f_p \sim p_{\text{sel}}(\cdot \mid \mathcal{P}_t), \quad p_{\text{sel}}(f \mid \mathcal{P}_t) \propto \phi(-\ell(f)), \quad (2)$$

where $\phi(\cdot)$ is a monotone transformation that assigns larger weights to lower-loss expressions. Common choices of the selection strategy include Top-$K$ selection, rank-based sampling, and Boltzmann sampling.

In the **variation** stage, the LLM proposes a new symbolic skeleton conditioned on the selected parent and its feedback:

$$\tilde{f} \sim p_{\theta}(\cdot \mid f_p, \ell(f_p), \mathcal{D}, Q), \quad (3)$$

where $p_{\theta}$ denotes the LLM and $Q$ denotes problem context. The skeleton $\tilde{f}$ specifies the discrete functional structure, including variables, operators, and their compositions, but may contain unknown numerical constants. Let $\mathbf{c}$ denote these constants and write the instantiated expression as $\tilde{f}_{\mathbf{c}}$. The constants are fitted by solving the following equation:

$$\mathbf{c}^{\star} = \arg\min_{\mathbf{c}} \ell(\tilde{f}_{\mathbf{c}}), \quad f_{\text{new}} = \tilde{f}_{\mathbf{c}^{\star}}. \quad (4)$$

This continuous optimization is commonly performed with the Broyden–Fletcher–Goldfarb–Shanno (BFGS) algorithm (Fletcher, 2013; Broyden, 1970; Fletcher, 1970; Shanno, 1970; Broyden, 1970). BFGS is a quasi-Newton method that approximates second-order curvature information without explicitly computing the Hessian. Starting from an initial constant vector, it repeatedly estimates a descent direction using an updated inverse-Hessian approximation and performs a line search to reduce the objective. This makes BFGS well-suited for the low-dimensional smooth optimization problems that arise when fitting constants after the symbolic skeleton is fixed.

In the **evaluation** stage, the completed candidate is scored on the dataset and incorporated into the population:

$$s_{\text{new}} = \ell(f_{\text{new}}), \quad \mathcal{P}_{t+1} = \text{Update}(\mathcal{P}_t, f_{\text{new}}, s_{\text{new}}). \quad (5)$$

The update rule typically retains high-fitness candidates while maintaining population diversity. After $T$ rounds, the best expression $\hat{f}$ in the final population $\mathcal{P}_T$ is returned:

$$\hat{f} = \arg\min_{f \in \mathcal{P}_T} \ell(f). \quad (6)$$

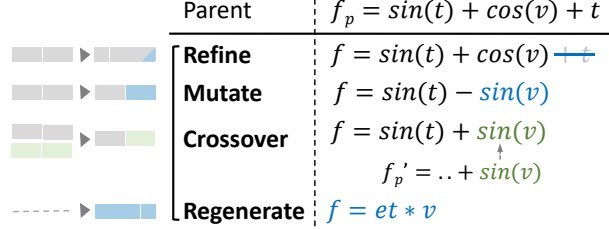

*Figure 3.* **Symbolic evolution operators.** The framework proposes new equations from a parent expression through four complementary operations: refinement for simplification, mutation for functional variation, crossover for reusing promising substructures, and regeneration for broader exploration.

## 3. Method

We present Deliberate Evolution (DE), an evolutionary framework for symbolic regression that makes LLM-based search more deliberate. The key principle is to decouple *candidate proposal* from *search guidance*: the LLM proposes symbolic skeletons, while explicit guidance determines how the search should proceed. Specifically, adaptive operators determine the refinement direction, analytical tools diagnose structural errors, and reflective memory summarizes useful experience from previous rounds.

As in Fig. 4, at round $t$, DE samples a parent expression $f_p$ from the population $\mathcal{P}_t$, selects an operator $o_t$ (Sec. 3.1), constructs a diagnostic report $a_t$ (Sec. 3.2), and retrieves memory $\mathcal{M}_{t-1}$ (Sec. 3.3). The LLM then proposes a skeleton $\tilde{f}_t \sim p_{\theta}(\cdot \mid Q, f_p, o_t, a_t, \mathcal{M}_{t-1})$, where $Q$ denotes the problem context. BFGS fits the constants in $\tilde{f}_t$ to obtain a complete expression $f_t$, which is evaluated, inserted into the population, and used to update both the operator policy and the memory. Algorithm 1 summarizes the full procedure.

### 3.1. Directional Guidance via Adaptive Operators

This module selects the operator $o_t$ used in the guided proposal distribution. The operator tells the LLM how to modify the selected parent expression $f_p$: refine it locally, perturb its structure, recombine it with another expression, or restart from a new hypothesis.

**Operator Set.** We define four semantic operators $\mathcal{O} = \{o_{\text{ref}}, o_{\text{mut}}, o_{\text{cross}}, o_{\text{reg}}\}$, ordered from exploitation to exploration. The *Refine* operator $o_{\text{ref}}$ makes conservative edits while preserving the parent skeleton. The *Mutate* operator $o_{\text{mut}}$ introduces structural changes, such as adding, removing, or replacing sub-expressions. The *Crossover* operator $o_{\text{cross}}$ recombines $f_p$ with an elite expression $f_p' \in \text{TopK}(\mathcal{P}_t)$. The *Regenerate* operator $o_{\text{reg}}$ ignores the parent structure and proposes a new expression from scratch. Examples of these operators are shown in Fig. 3.

**Selecting $o_t$.** Given a parent expression $f_p$, we first map it to a discrete search state $s_t$. The state is defined by two

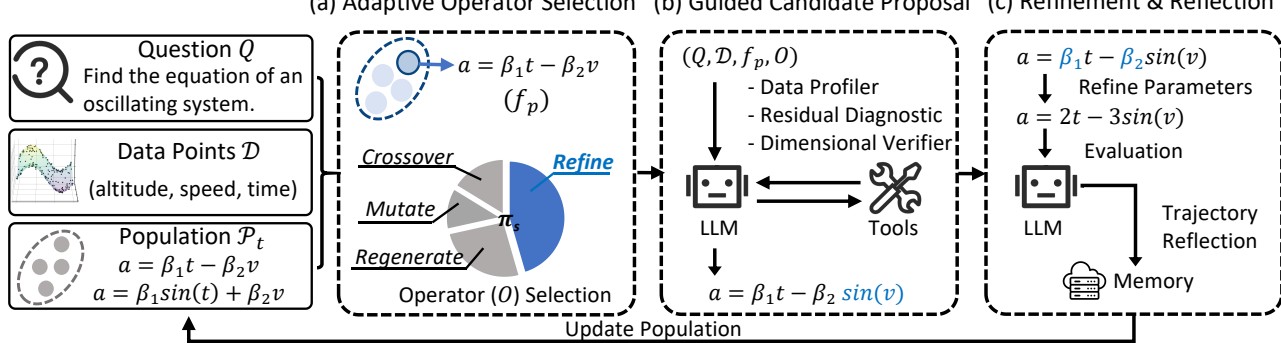

*Figure 4.* **Overview of Deliberate Evolution**. Deliberate Evolution follows a standard evolutionary loop, while explicitly decoupling LLM-based candidate proposal from deliberate guidance signals, including: (a) directional guidance via adaptive operator selection, (b) diagnostic guidance via tool-based analytical feedback, and (c) historical guidance via reflective memory distilled from trajectories.

binary signals: the normalized rank $\tilde{r}_p$, which measures expression quality, and the visit count $v_p$, which measures how often the corresponding search region has been explored:

$$s_t = S(f_p) = (\mathbb{I}[\tilde{r}_p \le \tau_r], \mathbb{I}[v_p \ge \tau_v]) \in \{0,1\}^2. \quad (7)$$

Here, $\tau_r$ and $\tau_v$ are thresholds for quality and maturity. Since $s_t$ contains two binary indicators, it induces four possible search states, corresponding to different combinations of candidate quality and exploration maturity. For each state $s \in \{0,1\}^2$, we maintain a nonnegative weight vector $\mathbf{w}_s^{(t)} = \{w_s^{(t)}(o) : o \in \mathcal{O}\}$ over operators. The operator policy $\boldsymbol{\pi}_s^{(t)}$ is a categorical distribution obtained by normalizing these weights with an exploration floor:

$$\boldsymbol{\pi}_s^{(t)}(o) = (1-\alpha)\frac{w_s^{(t)}(o)}{\sum_{o' \in \mathcal{O}} w_s^{(t)}(o')} + \frac{\alpha}{|\mathcal{O}|}. \quad (8)$$

The directional operator is sampled as $o_t \sim \boldsymbol{\pi}_s^{(t)}(\cdot)$. Thus, operator selection follows the chain $f_p \mapsto s_t \mapsto \boldsymbol{\pi}_s^{(t)} \mapsto o_t$: the parent determines the search state, the state indexes its categorical policy, and the policy samples the operator.

**Updating $\boldsymbol{\pi}_s^{(t)}(\cdot)$.** After applying $o_t$, the LLM proposes a skeleton, BFGS fits its constants, and the offspring $f_t$ is evaluated. We score the selected operator by the relative improvement over its parent:

$$r_t = \text{clip}\left(\frac{\ell(f_p) - \ell(f_t)}{\ell(f_p) + \varepsilon}, r_{\min}, r_{\max}\right). \quad (9)$$

Here, $r_t > 0$ means that $o_t$ improves the parent, while $r_t < 0$ means that it degrades performance. We update only the weight of the selected operator in state $s_t$:

$$w_s^{(t+1)}(o) = w_s^{(t)}(o) \cdot \begin{cases} \max(\delta, 1 + \eta r_t), & o = o_t, \\ 1, & o \ne o_t, \end{cases} \quad (10)$$

where $\eta$ controls the update strength and $\delta$ prevents excessive decay. The next policy is obtained by normalizing the

updated weights using the same exploration-floor rule. This update increases the future probability of operators that improve similar parents and decreases that of operators that fail, while keeping all operators available for exploration.

**Escaping Stagnation.** Although the state-based policy adapts operator choices locally, the population can still become trapped in a stagnant region. We monitor the best population loss $\ell_t^\star = \min_{f \in \mathcal{P}_t} \ell(f)$ and trigger stagnation control when no sufficiently large improvement occurs within a window of length $h$:

$$\text{Stag}_t = \mathbb{I}\left[\max_{j=t-h,\dots,t-1} \frac{\ell_j^\star - \ell_{j+1}^\star}{\ell_j^\star + \varepsilon} < \xi\right]. \quad (11)$$

When $\text{Stag}_t = 1$, DE temporarily shifts from exploitation to exploration: it relaxes parent selection, increases the probability of exploratory operators such as $o_{\text{mut}}$ and $o_{\text{reg}}$, and summarizes stagnated trajectories into memory for future avoidance. The operator policy therefore provides state-level adaptation, while stagnation control provides a global escape mechanism against repeated local edits.

### 3.2. Diagnostic Guidance via Tool-Augmented Proposal

This module constructs the diagnostic report $a_t$ used in the guided proposal distribution. Scalar feedback, such as MSE, measures how well a candidate fits the data, but it does not explain why the candidate fails. To provide localized evidence for revision, DE applies a diagnostic toolkit $\mathcal{T} = \{T_{\text{data}}, T_{\text{res}}, T_{\text{dim}}\}$ to the dataset, the parent expression, and the problem context. Specifically, given the selected parent $f_p$, the diagnostic report $a_t$ is computed as

$$a_t = \left[T_{\text{data}}(\mathcal{D}), T_{\text{res}}(f_p, \mathcal{D}), T_{\text{dim}}(f_p, Q)\right]. \quad (12)$$

As illustrated in Fig. 5, each tool captures a different source of structural information: (1) The *data profiler* $T_{\text{data}}$ summarizes properties of the data, including variable ranges, operator domains, interaction statistics, and patterns such as

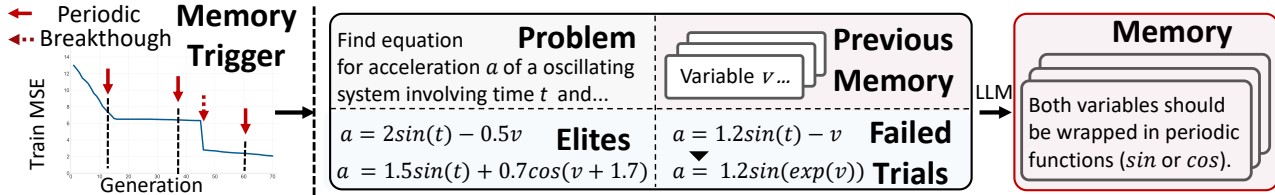

### (1) Data Profiler
Profile how input variables relate to the output.

Raw Input Data
$a = f^*(t, v)$

Project Variables

Projected View
a vs. t

### (2) Residual Diagnostic
Diagnose fitting errors from residual patterns.

Ground Truth
Model Estimate

Measure residuals

Target vs. Estimation

Ground Truth
Model Estimate

Residual Pattern

### (3) Dimensional Verifier
Verify unit compatibility.

kg

Inconsistency

cm

Mass: kg          Length: cm

*Figure 5.* **Diagnostic tools for symbolic evolution.** The tools profile raw input data, diagnose residual patterns, and verify dimensional consistency, providing actionable signals that guide later candidate generation toward valid and better-fitting symbolic expressions.

← Periodic
◄·· Breakthrough

**Memory Trigger**

Train MSE

Generation

**Problem**
Find equation for acceleration $a$ of a oscillating system involving time $t$ and...

$a = 2sin(t) - 0.5v$   **Elites**
$a = 1.5sin(t) + 0.7cos(v + 1.7)$

Variable $v$ ...

**Previous Memory**

$a = 1.2sin(t) - v$   **Failed**
$a = 1.2sin(exp(v))$   **Trials**

LLM

**Memory**

Both variables should be wrapped in periodic functions (*sin* or *cos*).

*Figure 6.* **Memory trigger for symbolic evolution.** When the search stagnates or reaches a periodic checkpoint, the LLM summarizes the problem, elite equations, previous memory, and failed trials into an actionable memory rule. The new memory then guides later candidate generation, helping the search escape plateaus and discover better symbolic expressions.

periodicity or singularity. (2) The *residual diagnostic* $T_{\text{res}}$ analyzes residuals $e_i(f_p) = y_i - f_p(\mathbf{x}_i)$ to identify systematic errors, such as missing terms or oscillatory patterns. (3) The *dimensional verifier* $T_{\text{dim}}$ checks physical-unit consistency and filters dimensionally invalid compositions when unit information is available. The resulting report $a_t$ is then injected into the context, enabling targeted revisions rather than undirected mutations.

### 3.3. Historical Guidance via Reflective Memory

This module updates the reflective memory $\mathcal{M}_t$, which stores reusable guidance distilled from past search trajectories. Let $\mathcal{H}_t = \{(f_p^{(j)}, o_j, f_j, \ell(f_j), r_j)\}_{j=1}^t$ denote the search history. Instead of updating memory at every round, DE updates it only when new information is likely to be useful: either periodically or after a significant improvement.

**Triggering Memory Update.** We decide whether memory should be updated at round $t$. Let $\ell_t^\star = \min_{f \in \mathcal{P}_t} \ell(f)$ denote the best loss, and let $\Delta_t = (\ell_{t-1}^\star - \ell_t^\star)/(\ell_{t-1}^\star + \varepsilon)$ be its relative improvement. The update trigger is

$$\delta_t = \mathbb{I}\left[(t \bmod K = 0) \vee (\Delta_t > \epsilon_{\text{mem}})\right], \quad (13)$$

where $K$ controls periodic updates and $\epsilon_{\text{mem}}$ controls breakthrough updates. Thus, $\delta_t = 1$ indicates that the current trajectory contains information worth reflecting on.

**Constructing the Reflection Context.** When $\delta_t = 1$, DE builds a structured context for reflection:

$$\mathcal{C}_t = \left(Q, \mathcal{M}_{t-1}, \text{Elite}(\mathcal{P}_t), \text{Fail}(\mathcal{H}_t), \text{Break}(\mathcal{H}_t)\right). \quad (14)$$

As illustrated in Fig. 6, $\text{Elite}(\mathcal{P}_t)$ contains high-performing

expressions, $\text{Fail}(\mathcal{H}_t)$ contains edits that substantially degrade their parents, and $\text{Break}(\mathcal{H}_t)$ contains edits that yield large improvements. Contrasting successful and unsuccessful patterns helps the LLM extract reusable lessons.

**Updating $\mathcal{M}_t$.** The LLM distills new insights from $\mathcal{C}_t$, and memory is updated only when the trigger is active:

$$\mathcal{M}_t = \begin{cases} \text{Compress}\left(\mathcal{M}_{t-1} \cup p_\theta(\cdot \mid \mathcal{C}_t)\right), & \delta_t = 1, \\ \mathcal{M}_{t-1}, & \delta_t = 0. \end{cases} \quad (15)$$

The compression step keeps memory concise by retaining recurring successful patterns, recording common failure modes, and removing redundant insights. The resulting $\mathcal{M}_t$ is used as historical guidance in subsequent proposal rounds.

### 3.4. Efficiency Analysis

We analyze the sample efficiency of DE from a local hitting-time perspective. This abstraction considers repeated proposals under a fixed parent $f_p$ and fixed guidance $g_t$, capturing the basic unit of evaluation in the evolutionary process. Let $\mathcal{F}^\star = \{f \in \mathcal{F} : \ell(f) \leq \lambda\}$ denote the set of target-quality expressions. For a fixed parent, let $q_0(\cdot \mid f_p)$ be an unguided proposal distribution and $q_\theta(\cdot \mid f_p, g_t)$ be the guided proposal distribution induced by DE, where $g_t = (o_t, a_t, \mathcal{M}_{t-1})$. The corresponding one-step success probabilities are

$$p_0 = \Pr_{f \sim q_0(\cdot|f_p)}[f \in \mathcal{F}^\star], \quad p_\theta = \Pr_{f \sim q_\theta(\cdot|f_p, g_t)}[f \in \mathcal{F}^\star]. \quad (16)$$

**Assumption.** We assume that explicit guidance improves the one-step success probability by a margin $\gamma > 0$, *i.e.*,

---

**Algorithm 1** Deliberate Evolution

---

**Require:** Dataset $\mathcal{D}$, problem context $Q$, budget $T$, toolkit $\mathcal{T}$, LLM $p_\theta$, state space $\mathcal{S}$

1: **Initialize:** population $\mathcal{P}_0$, memory $\mathcal{M}_0 \leftarrow \emptyset$, history $\mathcal{H}_0 \leftarrow \emptyset$, temperature $\tau$, weight vectors $\{\mathbf{w}_s^{(0)}\}_{s \in \mathcal{S}}$ and operator policies $\{\boldsymbol{\pi}_s^{(0)}\}_{s \in \mathcal{S}}$
2: **for** $t = 1$ **to** $T$ **do**
3:     Sample parent $f_p \sim \text{Boltzmann}(\mathcal{P}_{t-1}, \tau)$
4:     *# Directional guidance*
5:     $s_t \leftarrow S(f_p); \quad o_t \sim \boldsymbol{\pi}_{s_t}^{(t-1)}(\cdot)$
6:     *# Diagnostic guidance with tools*
7:     $a_t \leftarrow [T_{\text{data}}(\mathcal{D}), T_{\text{res}}(f_p, \mathcal{D}), T_{\text{dim}}(f_p, Q)]$
8:     $\tilde{f}_t \sim p_\theta(\cdot \mid Q, f_p, o_t, a_t, \mathcal{M}_{t-1})$
9:     $f_t \leftarrow \text{BFGS}(\tilde{f}_t, \mathcal{D})$
10:    $\mathcal{P}_t \leftarrow \text{Update}(\mathcal{P}_{t-1}, f_t)$
11:    *# Policy and memory update*
12:    Compute reward $r_t$ by Eq. 9
13:    Update $\mathbf{w}_{s_t}^{(t)}$ and $\boldsymbol{\pi}_{s_t}^{(t)}$ by Eq. 10
14:    $\mathcal{H}_t \leftarrow \mathcal{H}_{t-1} \cup \{(f_p, o_t, f_t, \ell(f_t), r_t)\}$
15:    Update memory $\mathcal{M}_t$ by Eq. 15
16:    *# Stagnation control*
17:    **if** $\text{Stag}_t = 1$ **then**
18:       Increase exploration in $\tau$ and $\{\boldsymbol{\pi}_s^{(t)}\}$
19:       $\mathcal{M}_t \leftarrow \mathcal{M}_t \cup \text{SummarizeStag}(\mathcal{P}_{t-h:t})$
20:    **end if**
21: **end for**
22: **Return** $\text{Best}(\mathcal{P}_T)$

---

$p_\theta \geq p_0 + \gamma$. Here, $\gamma$ captures the aggregate benefit of adaptive operators, diagnostic tools, and reflective memory. This assumption does not require guided proposals to be optimal; it only requires a positive improvement over unguided proposals, which may be arbitrarily small.

Let $N_\theta = \inf\{k \geq 1 : f_k \in \mathcal{F}^\star\}$ denote the hitting time to the target set under guided proposals. Assume conditionally independent sampling given fixed $(f_p, g_t)$, we have

$$\Pr(N_\theta > k) \leq (1 - p_\theta)^k \leq \exp[-k(p_0 + \gamma)], \quad \forall k \geq 1. \tag{17}$$

**Implication.** Eq. 17 shows that guided proposals yield a steeper exponential decay in failure probability than unguided proposals, whose failure probability decays as at most $\exp(-k p_0)$. Therefore, even a modest increase in one-step success probability can substantially reduce the number of evaluated candidates needed to reach a high-quality expression. See Appendix D for additional discussion.

# 4. Experiments

This section presents a comprehensive evaluation of Deliberate Evolution with representative baselines across various domains. We begin by introducing the experimental setups

(Sec. 4.1), followed by an analysis of the main performance (Sec. 4.2) and more detailed empirical analyses (Sec. 4.3).

## 4.1. Experimental Setup

**Benchmark.** We evaluate on LLM-SRBench (Shojaee et al., 2025b), a symbolic regression benchmark designed to reduce memorization of canonical formulas. It contains 240 problems from physics, chemistry, biology, and material science. Details are provided in Appendix E.1.

**Baselines.** We compare Deliberate Evolution with representative LLM-based symbolic regression methods, including LLMDirect, LLM-SR (Shojaee et al., 2025a), LASR (Grayeli et al., 2024), and SGA (Ma et al., 2024). Following LLM-SRBench, LLMDirect serves as a Best-of-$N$ prompting baseline. Details are provided in Appendix E.2.

**Metrics.** We report normalized mean squared error (NMSE) and accuracy under 1% relative tolerance ($\text{Acc}_{0.01}$). NMSE measures scale-normalized fitting quality, while $\text{Acc}_{0.01}$ measures the fraction of predictions that remain within a strict relative-error bound. Following prior protocols (Biggio et al., 2021; Kamienny et al., 2022), we discard the worst 5% predictions to reduce sensitivity to singular outliers:

$$\text{NMSE} = \frac{\sum_{i=1}^{N_{\text{test}}} (\hat{y}_i - y_i)^2}{\sum_{i=1}^{N_{\text{test}}} (y_i - \bar{y})^2},$$

$$\text{Acc}_{0.01} = \mathbb{I}\left(\max_i \left|\frac{\hat{y}_i - y_i}{y_i}\right| \leq 0.01\right), \tag{18}$$

where $\hat{y}_i$ is the prediction, $y_i$ is the ground truth, $\mathbb{I}(\cdot)$ is the indicator function, and $i$ indexes test samples.

**Implementation.** We run experiments with Llama-3.1-8B-Instruct (Dubey et al., 2024) and Qwen3-4B-Instruct-2507 (Yang et al., 2025) using vLLM (Kwon et al., 2023), with temperature set to 0.8. Following LLM-SRBench, baselines use a budget of 1,000 LLM-generated candidate expressions per problem; LASR further explores approximately $10^5$ non-LLM mutations. In contrast, Deliberate Evolution uses at most 400 samples per problem, corresponding to 40% of the standard LLM sampling budget. Additional details are provided in Appendix E.

## 4.2. Performance Analysis

**Deliberate Evolution improves both fitting accuracy and symbolic reliability.** Tab. 1 shows that Deliberate Evolution consistently achieves the best or competitive results across two models. On LSR-Transform with Qwen3-4B, Deliberate Evolution reduces NMSE from the strongest baseline of 1.83e-1 to 1.15e-1, while improving $\text{Acc}_{0.01}$ from 30.91% to 50.45%. This indicates that the method enhance both numerical fitting and strict relative-error accuracy.

**The improvement is consistent across scientific domains.**

*Table 1.* **Evaluation results on the LLM-SRBench benchmark** across LSR-Transform and LSR-Synth (Physics, Material Science, Chemistry, and Biology) using Llama3.1-8B-Instruct and Qwen3-4B-Instruct-2507. Metrics include NMSE ($\downarrow$) and Acc$_{0.01}$ ($\uparrow$, %). **Bold** and underlined values indicate the best and second-best performance per backbone model, respectively.

| Method | LSR-Transform | | Physics | | Material | | Chemistry | | Biology | |
| --- | --- | --- | --- | --- | --- | --- | --- | --- | --- | --- |
| | NMSE | Acc$_{0.01}$ | NMSE | Acc$_{0.01}$ | NMSE | Acc$_{0.01}$ | NMSE | Acc$_{0.01}$ | NMSE | Acc$_{0.01}$ |
| | | | | Llama3.1-8B-Instruct | | | | | | |
| **LLMDirect** | 2.95e-1 | 34.23 | 9.95e-3 | 0.00 | 8.16e-2 | 24.00 | 1.22e0 | 2.78 | 1.29e-1 | 4.17 |
| **LLM-SR** | 2.42e-1 | 34.23 | 3.00e-3 | 6.82 | 2.16e-1 | 60.00 | 5.24e-2 | **16.67** | 1.76e-2 | 12.50 |
| **LASR** | 2.62e-1 | 33.33 | 6.07e-3 | 9.09 | 9.47e-4 | 32.00 | 1.82e-3 | 8.33 | **6.40e-3** | 0.00 |
| **SGA** | 3.52e-1 | 0.90 | 1.55e-1 | 2.27 | 4.35e-2 | 12.00 | 4.58e-2 | 8.33 | 2.42e-1 | 0.00 |
| **Deliberate Evolution** | **1.12e-1** | **36.04** | **1.01e-3** | **11.36** | **2.89e-4** | **64.00** | **4.16e-4** | 11.11 | 1.17e-2 | **16.67** |
| | | | | Qwen3-4B-Instruct | | | | | | |
| **LLMDirect** | 3.55e-1 | 24.32 | 5.46e-2 | 6.82 | 1.42e-3 | 52.00 | 2.66e-1 | 2.78 | 4.46e-2 | 8.33 |
| **LLM-SR** | 3.15e-1 | 26.13 | 2.51e-3 | 6.82 | 3.55e-3 | 44.00 | 3.36e-2 | **13.89** | 1.88e-2 | **12.50** |
| **LASR** | 1.83e-1 | 30.91 | 6.04e-3 | 6.82 | 6.21e-4 | 8.00 | 2.31e-3 | 2.78 | 9.56e-3 | 0.00 |
| **SGA** | 4.09e-1 | 19.81 | 1.04e-1 | 0.00 | 1.02e-2 | 16.00 | 1.61e-1 | 2.78 | 1.73e-1 | 4.17 |
| **Deliberate Evolution** | **1.15e-1** | **50.45** | **4.37e-4** | **15.91** | **1.47e-4** | **56.00** | **1.88e-4** | **13.89** | **6.69e-3** | **12.50** |

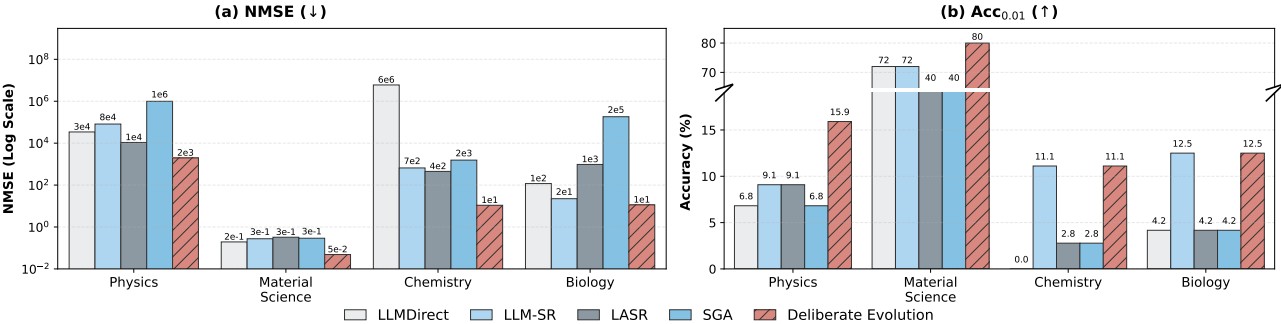

*Figure 7.* **Out-of-Distribution generalization on the LSR-Synth benchmark** across four scientific domains using Qwen3-4B. We report (a) NMSE (log scale) and (b) accuracy (Acc$_{0.01}$). The y-axis in (b) is segmented to accommodate scale differences.

On LSR-Synth with Qwen3-4B, Deliberate Evolution obtains the lowest NMSE on all four domains, including 4.37e-4 on Physics, 1.47e-4 on Material Science, 1.88e-4 on Chemistry, and 6.69e-3 on Biology. Similar gains are observed with Llama3.1-8B, suggesting that the method is not tailored to a single equation family or backbone model.

**Deliberate Evolution is particularly effective at precise expression refinement.** While strong baselines often remain at NMSE levels of $10^{-2}$ or $10^{-3}$, Deliberate Evolution frequently reaches the $10^{-4}$ range. For example, with Qwen3-4B, Deliberate Evolution improves the strongest baseline from 2.51e-3 to 4.37e-4 on Physics and from 2.31e-3 to 1.88e-4 on Chemistry. This suggests that structured guidance helps refine constants and local symbolic structures after promising expressions are found.

**The discovered expressions generalize better under distribution shift.** As shown in Fig. 7, Deliberate Evolution maintains much lower OOD NMSE than baselines across domains. Several baselines exhibit severe error amplification, exceeding 8e4 in Physics and 5e6 in chemistry, whereas Deliberate Evolution keeps errors at a much smaller scale, such as around 1e1 in chemistry. Together with the highest or

tied-best OOD Acc$_{0.01}$, these results suggest that Deliberate Evolution recovers more robust symbolic structures rather than merely fitting the training samples.

### 4.3. Robustness and Practical Validation

**Deliberate Evolution is stable across independent runs.** We evaluate run-to-run stability on the physics subset using three independent runs with Qwen3-4B and Llama3.1-8B. As shown in Fig. 8, Deliberate Evolution achieves the lowest variance while maintaining strong average performance across both backbones. With Qwen3-4B, Deliberate Evolution obtains a mean NMSE of 4e-4 with a variance of 9e-10; with Llama3.1-8B, it obtains a mean NMSE of 9e-4 with variance 6e-9. This suggests that adaptive operator selection, diagnostic tools, and reflective memory make the search less sensitive to stochastic variation.

**Deliberate Evolution is robust to noisy observations.** To simulate imperfect measurements, we inject Gaussian noise with $\sigma \in \{1\%, 5\%\}$ into the LSR-Transform training data using Qwen3-4B. As shown in Tab. 2, Deliberate Evolution achieves the lowest NMSE across all noise levels. At $\sigma = 1\%$, it obtains 1.79e-1 NMSE, compared with 2.46e-1 for

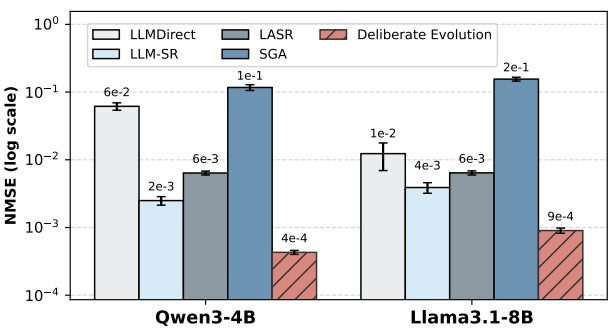

*Figure 8.* **NMSE (↓) on the Physics dataset across three independent runs.** Bars indicate the mean value, and error bars represent the minimum-maximum range.

*Table 2.* **NMSE (↓) results under different noise levels** on the LSR-Transform dataset using Qwen3-4B as backbone model.

| Method | Noise-Free | $\sigma = 1\%$ | $\sigma = 5\%$ |
|---|---|---|---|
| LLMDirect | 3.55e-1 | 4.29e-1 | 4.79e-1 |
| LLM-SR | 3.15e-1 | 4.01e-1 | 4.59e-1 |
| LASR | 1.83e-1 | 2.46e-1 | 2.70e-1 |
| **Deliberate Evolution** | **1.15e-1** | **1.79e-1** | **1.83e-1** |

*Table 3.* **Real-world symbolic regression results.** We report in-distribution (ID) and out-of-distribution (OOD) NMSE (↓).

| Method | ID-NMSE | OOD-NMSE |
|---|---|---|
| LLMDirect | 3.91e-1 | 1.20e0 |
| LLM-SR | 1.44e-1 | 6.34e-1 |
| LASR | 2.52e-1 | 1.15e0 |
| SGA | 3.95e0 | 1.84e0 |
| **Deliberate Evolution** | **1.11e-1** | **2.98e-1** |

*Table 4.* **Ablation results on the Physics dataset with Qwen3-4B.** We report NMSE (↓) and $Acc_{0.01}$ (↑, %).

| Part | Setting | NMSE | $Acc_{0.01}$ |
|---|---|---|---|
| **Full model** | Default | **4.37e-4** | **15.91** |
| *Component removal* | | | |
| Memory | w/o Memory | 1.34e-3 | 9.52 |
| Tool | w/o Tool | 2.52e-2 | 4.55 |
| *Operator selection* | | | |
| Operator | Fixed Refine | 8.69e-3 | 9.52 |
| Operator | Uniform | 1.02e-2 | 6.82 |
| Operator | w/o Stagnation | 7.69e-4 | 13.64 |

the strongest baseline. Its degradation from 1% to 5% noise is also smaller than that of the baselines, indicating that guided symbolic exploration favors more stable structural patterns rather than noise-sensitive fits.

**Deliberate Evolution remains effective on real-world measurements.** We further evaluate Deliberate Evolution on the Stress-Strain dataset (Aakash et al., 2019), a real-world symbolic regression task based on aluminum 6061-T651 measurements. Since this dataset contains measurement noise and imperfect symbolic correspondence, we report ID and OOD NMSE and omit $Acc_{0.01}$. As shown in Tab 3, Deliberate Evolution achieves the best ID and OOD performance, reducing ID-NMSE from 1.44e-1 to 1.11e-1 compared with LLM-SR and obtaining the lowest OOD-NMSE of 2.98e-1. These results suggest that the method can transfer beyond synthetic benchmark equations to noisy scientific measurements.

**Ablation Studies.** We isolate the contribution of reflective memory, diagnostic tools, and adaptive operator selection through controlled ablations. Reflective memory and diagnostic tools are removed, while adaptive operator selection is replaced by a fixed refine operator $o_{\text{ref}}$, uniform operator sampling, or a variant without the stagnation monitor. As shown in Tab. 4, every ablation degrades performance, indicating that the improvement does not come from a single component alone. Removing diagnostic tools causes the largest drop, increasing NMSE from 4.37e-4 to 2.52e-2 and reducing $Acc_{0.01}$ from 15.91% to 4.55%. Removing memory and simplifying operator selection also lead to clear degradation, showing that historical guidance, localized di-

agnosis, and adaptive exploration are complementary.

**Cross-Method and Tool-Guided Comparisons.** We provide qualitative comparisons to examine how structured guidance changes the search behavior. In Tab. 5, Deliberate Evolution recovers the correct or dominant symbolic skeletons, whereas baselines often replace missing structures with simpler surrogate terms. We also compare mutations from the same evolutionary state with and without diagnostic tools. Tool feedback identifies localized structural mismatches, such as missing periodic or polynomial components, and turns them into targeted edits with substantially lower MSE. These examples explain why diagnostic guidance improves symbolic recovery beyond scalar feedback.

## 5. Related Work

**Symbolic Regression (SR)** aims to identify mathematical expressions that characterize data. Conventional approaches fall into: (i) Search-based methods (Schmidt & Lipson, 2009; Sun et al., 2023), which explore equation space via stochastic operators. While effective, these methods suffer from high uncertainty, substantial computational inefficiency, and a tendency to generate bloated, uninterpretable expressions. (ii) Learning-based methods (Biggio et al., 2021; Kamienny et al., 2022), which employ trained neural models (*e.g.*, Transformers (Vaswani et al., 2017)) for direct prediction, yet remain constrained by heavy training data and limited generalization. Recently, (iii) LLM-based methods have emerged, leveraging encoded scientific knowledge in LLMs to propose expressions for iterative optimization (Shojaee et al., 2025a; Grayeli et al., 2024). However,

*Table 5.* **Qualitative case studies.** We omit fitted constants and compare symbolic skeletons for clarity. (a) Cross-method comparison of discovered equations. (b) Controlled comparison from the same evolutionary state with and without diagnostic tools.

| (a) Cross-method comparison of discovered symbolic skeletons. | | | |
|---|---|---|---|
| **Task (Domain)** | **Method** | **Expression** | **Structural note** |
| **P03** (Physics) | **Ground-truth** | $\sin(t) + \sin(v) + v$ | – |
| | LLMDirect | $\int v(t)\,dt + v + \cos(t) + \sin(t)$ | misses $\sin(v)$ |
| | LLM-SR | $\sin(t) + v + v^2 + v^3$ | polynomial surrogate |
| | LASR | $v + t$ | oversimplified |
| | SGA | $v + v^2$ | polynomial surrogate |
| | **Deliberate Evolution** | $\sin(\mathbf{t}) + \sin(\mathbf{v}) + \mathbf{v}$ | exact skeleton |
| **CRK4** (Chemistry) | **Ground-truth** | $A^2 + A\log(t+1)$ | – |
| | LLMDirect | $t + A + 1$ | oversimplified |
| | LLM-SR | $A + \frac{A^n}{1+(A/K)^n} + A^2 + tA + 1$ | extra rational terms |
| | LASR | $A^2 \sin\left(\left(e^t + \frac{1}{A}\right)^{-A}\right)$ | mismatched operator |
| | SGA | $A^2 + At + 1$ | linearizes $\log(t+1)$ |
| | **Deliberate Evolution** | $\mathbf{A^2} + \mathbf{A}\log(\mathbf{t+1}) + Ae^{-t}$ | dominant terms recovered |

| (b) Controlled comparison with and without diagnostic tools. | | | | |
|---|---|---|---|---|
| **Task** | **Setting** | **Expression** | **Diagnostic signal** | **MSE** |
| **P010** | Ground-truth | $\sin(t) + xt + x^3$ | residuals correlate with $\sin(t)$ (corr. $= -0.67$), may missing periodic term | – |
| | Parent | $xt + x^3$ | – | 4.4e-1 |
| | **w/ tools** | $\sin(\mathbf{t}) + \mathbf{xt} + \mathbf{x^3}$ | uses diagnostic signal | **7.8e-12** |
| | w/o tools | $xt + x^3 + x^2$ | unavailable | 1.2e-1 |
| **CRK3** | Ground-truth | $A^2 + A\exp(t) + \cos(\log(A))$ | parabolic residual pattern in $A$, which indicates missing $A^2$ term | – |
| | Parent | $A + t$ | – | 6.3e-4 |
| | **w/ tools** | $\mathbf{A^2} + \mathbf{A}\exp(\mathbf{t})$ | uses diagnostic signal | **2.5e-7** |
| | w/o tools | $A^3 + A + \sin(t)$ | unavailable | 9.7e-4 |

these methods typically rely on coarse scalar feedback, leading to notable sample inefficiency. In contrast, Deliberate Evolution steers the search via explicit, structured guidance signals, enhancing discovery quality at lower cost. We refer readers to Appendix B for a comprehensive discussion.

**Test-Time Scaling (TTS).** Scaling inference compute has proven effective for boosting LLM performance without parameter updates (Pang et al., 2025a). Existing TTS methods leverage either: (i) internal feedback (*e.g.*, Chain-of-Thought (Wei et al., 2022), Self-Consistency (Wang et al., 2023), and tree search (Yao et al., 2023)), relying on intrinsic model signals; or (ii) external feedback (*e.g.*, Best-of-N (Cobbe et al., 2021) and evolutionary search), utilizing environmental validation. In SR, where solution validity is strictly anchored to data fidelity, external feedback is indispensable. This renders evolutionary methods, which refine solutions via iterative external signals, a natural fit for SR. While recent frameworks (*e.g.*, AlphaEvolve (Novikov et al., 2025), FunSearch (Romera-Paredes et al., 2023), and ShinkaEvolve (Lange et al., 2025)) excel in general domains, they struggle with SR due to the tight coupling between discrete structural search and continuous constant refinement. To this end, we propose Deliberate Evolution, which enables strategic navigation through explicit, structured guidance.

## 6. Conclusion

This work introduces Deliberate Evolution, a novel framework that orchestrates candidate proposal with explicit guidance for symbolic regression. By integrating adaptive operators, analytical tools, and reflective memory, our approach promotes strategic exploration beyond trial-and-error optimization. Empirical evaluations across diverse domains show Deliberate Evolution outperforms baselines, achieving superior sample efficiency, generalization, and robustness. We encourage future research in symbolic regression for improved performance with reduced computational cost.

## Acknowledgment

This work received support from the National Science and Technology Major Project (No. 2022ZD0114903) and the Natural Science Foundation of China (NSFC. No. 62476149). ZKZ, XL, and BH were supported by NSFC Major Research Plan No. 92570109 and NSFC General Program No. 62376235. SC would also like to acknowledge the financial support from the Science and Technology Project of Beijing Municipal Science & Technology Commission (Grant No. Z251100008125030) and the Shuimu Scholar program from Tsinghua University.

## Impact Statement

This paper presents work whose goal is to advance the field of symbolic regression and agentic reasoning. There are many potential societal consequences of our work, none of which we feel must be specifically highlighted here.

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

# Appendix

# A. Further Discussions

## A.1. Limitations

**Scope of Backbone Models.** In this work, we evaluate our framework Deliberate Evolution on the LLM-SRBench dataset with Llama3.1-8B-Instruct and Qwen3-4B-Instruct-2507 as backbone models. Due to computational constraints, we have not conducted an exhaustive investigation with proprietary, closed-source models (*e.g.*, GPT, Gemini). While we believe the core findings are model-agnostic, the performance ceiling with stronger reasoners remains to be quantified. Although the experiments contain various scientific domains and various analyses, we lack the investigation of the closed-source models.

**Lack of Theoretical Convergence Guarantees.** Unlike traditional convex optimization or exhaustive search algorithms that may offer convergence proofs under specific conditions, our method relies on the stochastic generative nature of LLMs combined with heuristic evolutionary strategies. Consequently, we cannot theoretically guarantee that the global optimum will be found within a finite number of iterations. The framework operates as a probabilistic proposer-optimizer system, prioritizing efficient exploration of the functional space over guaranteed convergence.

**Inference Latency and Computational Cost.** Although Deliberate Evolution demonstrates high sample efficiency (*i.e.*, requiring fewer expression evaluations), the wall-clock time for each iteration is dominated by the inference latency of the LLM. Compared to primitive genetic operations in standard Genetic Programming (GP), the computational cost per step is inevitably higher. This currently limits the applicability of our method in real-time control scenarios or environments with extreme latency sensitivity.

## A.2. Future Directions

**Multi-Modal Evolutionary Frameworks.** Symbolic regression remains a challenging problem for LLMs, demanding a complex integration of domain expertise, data-informed search strategies, and mathematical reasoning capabilities. Effective perception of individual data points and their collective relationships is crucial for solving symbolic regression problems. Existing pure-text analytical tools often fall short of fully capturing the underlying data relationships, as they struggle to represent the holistic trends and patterns present in the data. In contrast, multimodal analysis tools can facilitate a more intuitive grasp of these global structures, especially those incorporating visual data representations such as plots and graphs. Future work could explore how multimodal large language models (MLLMs) can be leveraged to enhance symbolic regression, combining visual and symbolic reasoning to support more robust and interpretable model discovery.

**Scientific Discovery** Symbolic regression plays a fundamental role in scientific discovery, which requires theorem discovery and experimental verification. By enabling the automated extraction of interpretable mathematical expressions from empirical data, it serves as a powerful tool for AI-assisted research. Promising future directions include integrating symbolic regression with experimental design and physical simulations. Such advances could significantly accelerate scientific inquiry across various scientific domains.

**Robust Symbolic Regression under Noisy Conditions** Existing symbolic regression benchmarks typically assume that training data are accurately sampled from the underlying ground-truth equation. However, in real-world scenarios, sampling noise is often unavoidable. Consequently, identifying the correct equation from noisy data becomes a crucial and practical research challenge. Recovering equations from noisy data entails disentangling noise from the underlying signal and identifying the correct functional form, which remains a fundamentally challenging problem. Although we investigate the experimental performance under Gaussian noise, it is still underexplored for real stochastic noise or real-world conditions.

**Test-Time Training and Online Adaptation.** Currently, our framework utilizes LLMs in a frozen state, relying solely on test-time In-Context Learning (ICL) to adapt to new datasets. However, the internal weights of the LLM are not updated to capture the specific characteristics of the target problem instance. A compelling direction for future work is to incorporate Test-Time Training (TTT) or online fine-tuning strategies. By updating the model parameters (or lightweight adapters) on the validation signal of the test data during the evolutionary search, the "Proposer" can gradually overfit the specific domain syntax and structural patterns of the problem at hand, potentially leading to higher search efficiency and accuracy.

# B. Related Work

In this section, we provide a detailed extension of the related work discussed in Sec. 5, including (i) symbolic regression, (ii) test-time scaling methods, and (iii) evolutionary methods.

## B.1. Symbolic Regression

Symbolic Regression (SR) seeks to uncover symbolic mathematical expressions from observational data. SR simultaneously searches for both the functional structure (the arrangement of operators and variables) and the numerical constants. The search space $\mathcal{F}$ of SR is defined by a set of terminal nodes (variables and constants) and a set of primitive operators. A candidate solution is typically represented as a computational tree or a mathematical string.

**Fundamental Challenges.** The inherent difficulty of symbolic regression stems from several factors that make it significantly more challenging than standard optimization tasks:

- Combinatorial Explosion: The size of the search space $\mathcal{F}$ grows exponentially with the depth of the expression tree and the cardinality of the primitive operator set $|\mathcal{PO}|$. For a tree with $k$ nodes, there are roughly $O(|\mathcal{PO} \cup \mathcal{T}|^k)$ possible structures, making exhaustive search computationally intractable. Even for small $k$, the space is too vast for brute-force enumeration.

- Non-Differentiable Search Space: The structure of each expression $f$ is discrete, *i.e.*, small changes in the expression (*e.g.*, changing a $+$ to a $\times$) can lead to catastrophic changes in the output, creating a "rugged" fitness landscape. This prevents the direct use of gradient-based optimization for the structure.

- Accuracy-Parsimony Trade-Off: There is often a conflict between fitting the noise in the data and maintaining a simple, generalizable formula. Identifying the "elbow point" on the Pareto frontier—where an increase in complexity no longer yields significant gains in accuracy—remains a non-trivial model selection challenge.

**Conventional Methods.** Conventional SR approaches were primarily driven by two dominant paradigms, each with distinct strengths and inherent limitations:

- **Search-based Methods:** This paradigm treats SR as a discrete optimization problem, exploring the vast space of mathematical expressions through iterative search.
  - *Evolutionary Algorithms (EA):* Classic approaches like Genetic Programming (GP) (Schmidt & Lipson, 2009) evolve a population of expression trees via random crossover and mutation. These methods stochastically conduct mutations. While powerful at discovering complex non-linear relations, they are often criticized for the bloat phenomenon, where expressions grow unnecessarily large without improving fitness.
  - *Reinforcement Learning (RL):* Several advances (Petersen et al., 2021) try to utilize RNNs as policies to sample expressions, optimized via risk-seeking policy gradients. These methods suffer from low computational efficiency, high variance in convergence, and often require expert-level hyperparameter tuning to navigate the rugged fitness landscape.

- **Learning-based Methods:** Inspired by the success of neural networks, these methods frame SR as a supervised sequence-to-sequence task. They typically leverage transformers trained on large-scale synthetic data to predict equations directly (Biggio et al., 2021; Kamienny et al., 2022). While enabling rapid inference, these end-to-end models require substantial amounts of training data and often exhibit limited generalization.

**LLM-Based Methods.** The emergence of large language models (LLMs) has catalyzed a third paradigm: LLM-based symbolic regression methods. Unlike previous approaches, these methods treat LLMs as informed optimization agents that are capable of leveraging vast encoded scientific knowledge to constrain the search space (Shojaee et al., 2025a; Grayeli et al., 2024). Existing works implement LLMs within iterative frameworks, such as sequential refinement or evolutionary optimization. In these methods, LLMs typically act as a monolithic proposer, generating offspring expressions based on parent candidates and simple scalar feedback (*e.g.*, Mean Squared Error).

However, these methods generally rely on the potential of models without explicit guidance, forcing the models to engage in a stochastic trial-and-error process. This often leads to suboptimal sample efficiency and a high dependency on the model's inherent probabilistic sampling. In contrast, Deliberate Evolution steers the optimization process with explicit, structured guidance, including directional, diagnostic, and historical signals. By integrating these signals into the model-based generation loop, our framework enables a strategic, informed exploration. Consequently, Deliberate Evolution significantly reduces the computational sample budget required for discovery and achieves superior performance across various scientific domains.

## B.2. Test-Time Scaling Methods

The performance of LLMs can be significantly augmented by scaling inference-time computation (Lv et al., 2026), a paradigm often referred to as Test-Time Scaling (TTS). Unlike training-time scaling, TTS enhances model outputs without further parameter updates. TTS aims to trade computation (samples, time, or search) for better performance (Zhou et al., 2024). We roughly categorize these methods based on their feedback mechanisms and exploration strategies:

- **Internal Feedback:** This category relies on the model's intrinsic capacity to decompose problems and verify its own logic without requiring feedback from the environment (Zhou et al., 2025b). Recent studies highlight that LLMs' reasoning capabilities can be effectively elicited through carefully designed prompts. Methods such as Chain-of-Thought (CoT) (Wei et al., 2022) guide LLMs to produce step-by-step reasoning, enhancing performance on complex reasoning tasks. Similarly, Least-to-Most (LtM) prompting (Zhou et al., 2023) employs a divide-and-conquer strategy, breaking intricate problems into manageable sub-questions for improved problem-solving capacity. Also, prior works show that LLMs demonstrate reflection ability for self-improving performance in the absence of external feedback. For example, Self-Refine (Madaan et al., 2023) encourages LLMs to refine the initial outputs using their inherent reasoning capabilities. Also, Self-Consistency (SC) (Wang et al., 2023) generates multiple independent samples for a given problem and selects the most consistent answer through majority voting.

- **External Feedback:** Without external feedback from the environment may lead to bias due to models' inherent capability for hallucination. Therefore, it is important to provide valid external feedback signals from the environment for LLMs. Existing works show that LLMs can improve and refine their initial performance using external feedback (Pang et al., 2025b). For instance, Reflexion (Shinn et al., 2023) and Recursively Criticizes and Improves (RCI) (Kim et al., 2023) incorporate external feedback to iteratively validate and enhance solutions, further improving reliability. Tree-of-Thought (ToT) (Yao et al., 2023) or Graph-of-Thought (GoT) (Besta et al., 2024) methods further extend the CoT reasoning chains for improved performance. They decompose complex problems into multiple reasoning steps, exploring each step by sampling diverse reasoning paths and employing verifiers to identify the most promising solutions. Additionally, Monte Carlo Tree Search (MCTS) (DeLorenzo et al., 2024) enhances exploration by incorporating simulation and backpropagation within the tree-search process. Evolutionary methods maintain a population of candidate samples, which are iteratively optimized. We discuss Evolutionary methods in detail.

## B.3. Evolutionary Methods

Evolutionary Algorithms (EAs) provide a robust framework for global optimization in non-convex and non-differentiable spaces (Li et al., 2026a). By maintaining a population of candidate solutions and iteratively applying selection, crossover, and mutation, EAs can effectively navigate complex landscapes that are challenging for gradient-based methods. Recently, the integration of Large Language Models (LLMs) as "neural operators" within the evolutionary loop has defined a new frontier in automated discovery.

Existing works show that the synergy between LLMs and EAs has yielded significant breakthroughs in several domains (Li et al., 2026b), such as code generation (Romera-Paredes et al., 2023) and algorithm evolution (Novikov et al., 2025; Lange et al., 2025). While the aforementioned frameworks excel in general domains, their direct migration to Symbolic Regression is non-trivial. The difficulty of SR is rooted in its nature as a hybrid optimization problem. Unlike mathematical reasoning or code generation — where a solution might be partially correct through internal logic — an SR solution's validity is strictly anchored to its alignment with numerical data. This renders the feedback loop "zero-sum": a single incorrect operator can result in a catastrophic loss in fitness, even if the rest of the structure is promising. Consequently, external feedback is not just an evaluator but an indispensable compass for survival in the evolutionary process.

Existing LLM-based evolutionary methods for SR often treat the model as a stochastic proposer, relying on its probabilistic nature to "blind" trial-and-error via scalar feedback (*e.g.*, MSE). This approach is highly sample-inefficient, as it fails to exploit the diagnostic information latent in the data-model mismatch. This paradigm also suffers from collapsed exploration due to the inherent bias in LLMs (Zhou et al., 2026).

In contrast, Deliberate Evolution transforms symbolic regression into a deliberately guided discovery process. Instead of treating the LLM as a black-box optimizer, we leverage the LLM's reasoning capabilities to interpret explicit, structured guidance signals. By providing the model with directional, diagnostic, and historical insights, we enable a more informed evolution, leading to superior discovery quality and efficiency.

# C. Algorithmic Elaborations

In this section, we provide a comprehensive technical exposition of Deliberate Evolution, expanding upon the methodological framework introduced in Sec. 3. While the main text outlines the core philosophy of our approach, the following subsections delve into the granular algorithmic nuances and implementation specifics.

We specifically detail the generation of structured guidance signals, the mechanics of the LLM-driven evolutionary loop, and the specialized hybrid optimization protocols used to bridge discrete structural search with continuous constant refinement. Our goal is to provide a transparent and reproducible road-map that elucidates how Deliberate Evolution effectively navigates the complex search space of symbolic regression.

## C.1. Parent Expression Sampling

The selection mechanism in Deliberate Evolution is designed to balance exploitation (refining high-performing candidates) and exploration (maintaining population diversity). Unlike vanilla evolutionary algorithms that rely solely on fitness scores, our sampling strategy incorporates historical metadata and structural constraints to prevent premature convergence.

**Population Management.** Our framework maintains an archive of candidate expressions (*i.e.*, population $\mathcal{P}$). To ensure high genotypic diversity and prevent the search from collapsing into local functional optima, Deliberate Evolution employs a Multi-Island Experience Buffer: the population $\mathcal{P}$ is partitioned into several independent subsets, referred to as "islands". Each island evolves its own set of candidates, and occasional migration occurs between islands. This architecture prevents a single dominant but suboptimal functional form from colonizing the entire population.

To optimize the search quality, we implement two critical management policies:

- **Deduplication**: When a new candidate $f_{\text{new}}$ is registered, we check for symbolic equivalence within its island. If $f_{\text{new}}$ already exists, it replaces the incumbent only if it achieves a lower MSE (due to better constant refinement), ensuring functional diversity.

- **Island Reset**: To prevent stagnation, we implement a periodic reset mechanism. Every $G_{reset}$ generations, the bottom 50% of islands are cleared. These "weak" islands are then randomly sampled from the best candidates of the top-performing islands to re-initiate exploration from a promising functional neighborhood.

Each candidate expression $f$ in the population is stored with its symbolic string, optimized parameters, and metadata (including its mean squared error, the number of times it has been selected as a parent, and the performance of its best offspring).

**Fitness Definition.** Instead of relying solely on MSE, we define a composite fitness score $s_{\text{total}}$ that encapsulates accuracy, search efficiency, and potential. This multi-objective signal prevents the search from "greedily" following the lowest MSE and prioritizes candidates that are likely to lead to structural breakthroughs.

$$s_{\text{total}} = w_{\text{MSE}} \cdot s_{\text{MSE}} + w_{\text{Hist}} \cdot s_{\text{Hist}} + w_{\text{Comp}} \cdot s_{\text{Comp}}. \tag{19}$$

(i) Accuracy Component $s_{\text{MSE}}$: To mitigate the impact of extreme MSE outliers and provide a smooth selection pressure, we use a rank-based exponential decay. For a candidate with rank $rank$ within the population os size $N$ ((where $r = 0$ is the best)):

$$s_{\text{MSE}} = \exp(-\frac{rank/N}{\tau_{\text{rank}}}), \tag{20}$$

where $\tau_{\text{rank}}$ is a decay factor. This ensures that the top-performing candidates maintain a significant sampling advantage without completely zeroing out the middle-tier candidates.

(ii) Historical Component $s_{\text{Hist}}$: This term acts as a dynamic regulator, rewarding underexplored candidates and those demonstrating high "fertility" (ability to produce better offspring):

$$s_{\text{Hist}} = w_{\text{sample}} \cdot F_{\text{sample}} + w_{\text{elite}} \cdot F_{\text{elite}} \tag{21}$$

- Sampling Penalty ($F_{\text{sample}}$): To avoid redundant queries to the LLM for the same parent, we penalize candidates that have already been sampled $c_{\text{elite}}$ times:

$$F_{\text{sample}} = \exp(-\frac{c_{\text{elite}}}{\tau_{\text{elite}}}) \tag{22}$$

- Improvement Potential ($F_{\text{elite}}$): We reward parents that have successfully produced superior offspring, signaling that the current structural neighborhood is promising:

$$F_{\text{elite}} = \max(0, \frac{MSE_{\text{parent}} - MSE_{\text{best\_child}}}{MSE_{\text{parent}} + \epsilon}) \tag{23}$$

(iii) Complexity Component ($s_{\text{Comp}}$): This term is designed to incorporate Occam's Razor by penalizing excessively long or nested expressions, thereby steering the search toward the Pareto frontier of accuracy and parsimony.

**Parent Sampling.** The parent is sampled hierarchically following previous works: we first uniformly sample an island, and then sample parent expression within the island via Boltzmann Sampling over the composite fitness distribution. The probability of selection of candidate $s$ is given by:

$$P(f) = \frac{\exp(s_{\text{total}}(f)/T)}{\sum_{f' \in \mathcal{P}} \exp(s_{\text{total}}(f')/T)} \tag{24}$$

where $T$ is the Boltzmann temperature parameter. We employ a simulated annealing schedule, decreasing $T$ linearly or exponentially over generations to shift from broad exploration to focused refinement.

### C.2. Adaptive Operator Selection

This section provides the formal details of the adaptive operator selection in Deliberate Evolution, encompassing the semantic definitions of evolutionary operators and the optimization of the Multi-Armed Bandit (MAB) policy.

We introduce a set of four evolutionary operators $\mathcal{O} = \{o_{\text{ref}}, o_{\text{mut}}, o_{\text{cross}}, o_{\text{reg}}\}$ that span increasing degrees of modification to the parent. To facilitate understanding, we provide an example in Fig. 3.

To ensure the MAB remains stable under the high variance of LLM outputs, we implement several safety-critical mechanisms in the reward calculation and policy update phases.

- Robust Reward Shaping: The reward signal $r_t$ is designed to be both sensitive to structural improvements and resilient to catastrophic failures. A significant challenge in LLM-based evolution is the generation of syntactically invalid or non-executable expressions. We assign a fixed penalty reward of $r = -0.5$ for any operator that fails to produce a valid candidate. Also, to prevent a single "lucky" discovery from dominating the policy, we clip the reward to the range $[r_{\min}, r_{\max}]$, where $r_{\min} = -0.5$ and $r_{\max} = 1.0$. This ensures that even a perfect discovery ($MSE = 0$) provides a bounded update signal.

- Asymmetric Multiplicative Updates: The policy $\boldsymbol{\pi}_s$ for each state $s \in \mathcal{S}$ is updated using an asymmetric rule to handle positive and negative feedback differently. When an operator $o$ yields an improvement ($r > 0$), its probability is scaled by a factor related to the learning rate $\eta$:

$$P_{\text{new}}(o) = P_{\text{old}}(o) \cdot (1 + \eta \cdot \min(r, \tau_{\text{clip}})) \tag{25}$$

where $\tau_{\text{clip}}$ (default 2.0) prevents extreme probability jumps. When an operator fails or regresses ($r \leq 0$), we apply a decay factor to its selection probability:

$$P_{\text{new}}(o) = P_{\text{old}}(o) \cdot \max(\delta, 1 - \eta \cdot |r|) \tag{26}$$

The parameter $\delta$ (min decay factor, default 0.1) serves as a safety buffer, preventing an operator's weight from collapsing too rapidly due to the inherent stochasticity of LLM sampling.

- Exploration via Mixture Distributions: To guarantee continuous exploration in MAB, we do not use the raw normalized weights. Instead, we compute the final policy as a mixture of the learned distribution and a uniform distribution:

$$P_{\text{final}} = (1 - \epsilon) \cdot P_{\text{normalized}} + \epsilon \cdot P_{\text{uniform}} \tag{27}$$

where $\epsilon = \min(1.0, |\mathcal{O}| \cdot p_{\min})$ and $p_{\min}$ is the minimum probability floor (*e.g.*, 0.05). This mechanism ensures that every operator in the portfolio $\mathcal{O}$ retains a minimum sampling probability, allowing the bandit to rediscover the utility of previously "bad" operators if the search enters a new functional regime.

## C.3. Reflective Memory

Building upon the framework established in Sec. 3.3, this section elaborates on the storage and maintenance of the reflective memory $\mathcal{M}_t$. As illustrated in Fig. 6, when an update is triggered, the system distills raw trajectories into concise natural language insights. These insights are injected into the prompt of the candidate generation stage.

To ensure computational efficiency and prevent context length saturation in the LLM's input, we implement a strict capacity constraint on the memory module. We utilize a First-In-First-Out (FIFO) eviction policy: once the number of stored insights reaches the predefined threshold $N_{max}$, the oldest insights, which may reflect outdated search states, are removed to make room for fresh observations. This mechanism ensures that the LLM is always guided by the most recent and relevant discoveries, maintaining a lean yet potent historical context throughout the evolutionary process.

## C.4. Tool Design

This section extends the introduction of analytical tools described in Sec. 3.2. To shift the model from blind guessing to deliberate analysis (Zhou et al., 2025a), we introduce a toolkit of scientifically aware utilities. A detailed description of each tool's role and application is as follows.

**Design Principle.** Instead of feeding raw data to the LLM, which is often prone to numerical hallucination, our tools perform statistical tests and symbolic audits to extract: (i) Directional Cues: Suggesting specific operators or transformations based on correlation. (ii) Structural Constraints: Pruning the search space by enforcing mathematical and physical consistency. (iii) Error Attribution: Identifying precisely where and why a current hypothesis fails to capture the data distribution.

**Tool-1: Data Profiler** ($T_{\mathbf{data}}$). This tool analyses the data to propose structural hypotheses.

- Domain and Integrity Checks: Identifies variables that cross zero to avoid invalid $\log$ or $\sqrt{x}$ operations, and detects potential singularities by monitoring the kurtosis of the output distribution (e.g., $K > 10.0$ suggests resonance or poles).

- Uni-variate Feature Scanning: Computes Pearson and Spearman correlations across various transformations (Linear, Squared, $\sin$, $\cos$, $\exp$, $\log$). It flags non-linear but monotonic relationships where the Spearman rank exceeds the Pearson coefficient.

- Interaction and Structural Testing: When $|\mathbf{x}| \geq 2$, it tests variable products to detect coupling. For wide-range data (span $> 10^3$), it evaluates global structural archetypes, such as exponential growth ($\log |y| \sim x$) vs. rational decay ($1/y \sim x$).

**Tool-2: Residual Diagnostic** ($T_{\mathbf{res}}$). This tool provides corrective feedback by analyzing the unexplained variance (*i.e.*, residuals) of a candidate expression $f_p$.

- Missing Term Detection: It calculates the correlation between the residuals and all input variables under various transcendental transformations. For instance, a high correlation with $\sin(x)$ or $\exp(-x)$ suggests the current model lacks specific periodic or decay components.

- Heteroscedasticity Analysis: By examining the relationship between the magnitude of residuals and the predicted values $\hat{y}$ via Spearman's rank test, the tool identifies error patterns. For instance, if errors scale with $\hat{y}$, it recommends a rational or logarithmic functional form to stabilize the variance.

- Interaction Scanning: It explicitly checks if the residuals are correlated with variable couplings, providing the LLM with a clear signal to move beyond additive models.

**Tool-3: Dimensional Verifier** ($T_{\mathbf{dim}}$). To ensure physical plausibility, this tool performs recursive dimensional analysis on the symbolic expression tree using SymPy. Starting from base variable units, the tool recursively computes the dimensions of each node. It enforces strict Additive Consistency (terms in $A + B$ must have identical units) and Multiplicative Invariants.

## C.5. Optimization and Evaluation

This section details the process of transforming raw LLM-generated strings into numerically grounded and validated mathematical hypotheses.

**Numerical Optimization.** Following the established paradigms, we task the LLM exclusively with generating the functional skeleton of the expression. Given that LLMs exhibit inherent limitations in high-precision numerical reasoning, they are instructed to use generic placeholders for constants. For the generated skeleton, we perform numerical grounding by optimizing these constants using the BFGS (Broyden-Fletcher-Goldfarb-Shanno) algorithm. This hybrid approach decouples discrete structural search from continuous parameter optimization, leveraging the LLM's symbolic prior and BFGS's local convergence efficiency.

Prior works often encourage LLMs to generate executable Python functions. However, we observe that in this approach, minor formatting errors (*e.g.*, indentation or syntax mismatches) frequently lead to execution failures. To enhance system robustness, we require the LLM to output mathematical infix expressions. These strings are then parsed into a symbolic computation tree using Python packages such as SymPy. This abstraction layer allows us to perform automated structural simplification, derivative calculation, and robust error handling before numerical evaluation.

**Rejection Sampling.** To maintain the integrity of the search space, each candidate undergoes a rigorous structural audit. We implement Rejection Sampling based on variable consistency: if the LLM generates an expression containing independent variables not specified in the problem statement (*i.e.*, out-of-vocabulary variables), the sample is discarded immediately. This prevents the model from introducing spurious dependencies that could mislead the evolutionary trajectory.

**Data Splitting for Reliable Evaluation.** To prevent overfitting during the discovery process, we further partition the provided training data into two disjoint subsets:

- *Sub-training Set*: Used by the BFGS optimizer to find the optimal values for the symbolic constants.

- *Internal Validation Set*: Used to compute the fitness scores (MSE) for the candidate selection and memory updates.

This split ensures that the evaluation of a functional structure is not biased by the specific data points used to tune its constants, thereby promoting the discovery of truly generalizable laws.

**Population Registration.** Once evaluated, a new candidate is registered back into the population. Following the principle of previous studies, the offspring is added to the same island as its parent. This maintains the independent evolutionary trajectory of each island.

# D. Additional Efficiency Analysis

In this section, we provide the formal justifications for the modeling assumptions and present the detailed derivation of the hitting-time bounds.

## D.1. Preliminaries and Notation

We briefly recall the problem setup and notation from the main text to ensure this appendix is self-contained.

- **Search Space:** Let $\mathcal{F}$ be the space of all symbolic expressions defined by a predefined set of mathematical operators, variables, and constants. Let $\mathcal{F}^{\star} \subset \mathcal{F}$ denote the target set of expressions that satisfy a specific performance criterion, *e.g.*, $\mathrm{MSE} < \epsilon_{\mathrm{thre}}$.

- **Proposal Distributions:**

    - $q_0(\cdot \mid f_p)$: The baseline (unguided) proposal distribution given a parent $f_p$.
    - $q_\theta(\cdot \mid f_p, g_t)$: The guided proposal distribution conditioned on the guidance signal $g_t = (o_t, a_t \mathcal{M}_{t-1},)$.

- **Success Probabilities:** Conditioned on the parent $f_p$ and guidance signals $g_t$, we define the single-step success probabilities as:

$$p_0 := \Pr_{f \sim q_0} (f \in \mathcal{F}^{\star}), \quad p_\theta := \Pr_{f \sim q_\theta} (f \in \mathcal{F}^{\star}). \tag{28}$$

- **Hitting Time:** Let $N_0$ and $N_\theta$ be the random variables denoting the number of independent proposals required to sample the first $f \in \mathcal{F}^\star$ under the baseline and guided distributions, respectively.

## D.2. Justification of Assumptions

Our analysis relies on two key assumptions: the conditional independence of proposals and the informativeness of guidance. We justify them below.

**1. The Conditional Independence Assumption (Local Bernoulli Model).** Our analysis focuses on the sample complexity within a single evolutionary iteration. We consider the generation of a batch of $k$ candidates conditioning on a fixed parent expression $f_p$ and a fixed guidance signal $g_t$. Under this conditioning, each proposal is generated from the LLM using an identical prompt and decoding distribution (with fixed temperature), without modifying the internal parameters of the model of the prompt context between trials in the same batch.

Consequently, the sequence of candidate proposals $\{f_1, \ldots, f_k\}$ can be rigorously modeled as independent and identically distributed (i.i.d.) samples from the same distribution. This allows the proposal process to be viewed as a sequence of Bernoulli trials, which is standard in local sample complexity analyses and does not require long-horizon cross-iteration independence.

**2. The Guidance Advantage Assumption.** We assume that conditioning on guidance yields a strictly positive advantage $\gamma$, *i.e.*, $p_\theta \geq p_0 + \gamma$. This assumption is grounded in the explicit design of our guidance modules.

Mathematically, we can decompose the probability of generating a target expression as:

$$\Pr(f \in \mathcal{F}^\star) = \Pr(f \in \mathcal{F}^\star \mid f \in \mathcal{F}_{\text{valid}}) \cdot \Pr(f \in \mathcal{F}_{\text{valid}}), \tag{29}$$

where $\mathcal{F}_{\text{valid}} \subset \mathcal{F}$ denotes the subset of expressions that satisfy basic validity constraints (*e.g.*, dimensional consistency). Our guidance mechanisms improve these terms specifically:

1. **Pruning via Tools ($\mathcal{T}$):** The diagnostic tools act as a soft filter, significantly increasing $\Pr(f \in \mathcal{F}_{\text{valid}})$ by suppressing dimensionally or structurally invalid candidates.

2. **Biasing via Operators ($o_t, \mathcal{M}_t$):** The directional operators and reflective memory serve as priors that restrict the search to high-potential subspaces, thereby increasing the conditional probability $\Pr(f \in \mathcal{F}^\star \mid f \in \mathcal{F}_{\text{valid}})$.

The simultaneous improvement of these terms justifies the existence of an advantage margin $\gamma > 0$.

## D.3. Derivation of Hitting-Time Bounds

Here, we provide the step-by-step derivation of the exponential decay bound reported in the main text.

**Hitting Time Distribution.** Recall that $N_\theta := \inf\{k \geq 1 : f_k \in \mathcal{F}^\star\}$. Based on the conditional independence justified above, the hitting-time $N_\theta$ follows a geometric distribution with parameter $p_\theta$. The probability of failing to find a solution within a budget of $k$ trials is:

$$\Pr(N_\theta > k) = (1 - p_\theta)^k. \tag{30}$$

**Exponential Decay Bound.** We utilize the inequality $1 - x \leq e^{-x}$ (valid for all $x \in \mathbb{R}$). Applying this to the failure probability:

$$\Pr(N_\theta > k) \leq (e^{-p_\theta})^k = \exp(-k p_\theta). \tag{31}$$

Substituting the Guidance Advantage Assumption ($p_\theta \geq p_0 + \gamma$):

$$\Pr(N_\theta > k) \leq \exp\big(-k(p_0 + \gamma)\big). \tag{32}$$

This establishes the failure probability decays exponentially with the number of samples, with the decay rate accelerated by $\gamma$.

**Comparison with Unguided Baseline.** For the unguided baseline, an identical argument yields $\Pr(N_0 > k) \leq \exp(-kp_0)$. To quantify the acceleration, we consider the ratio of the failure probabilities:

$$\frac{\Pr(N_\theta > k)}{\Pr(N_0 > k)} = \left(\frac{1 - p_\theta}{1 - p_0}\right)^k \leq \left(1 - \frac{\gamma}{1 - p_0}\right)^k \leq \exp\left(-\frac{k\gamma}{1 - p_0}\right). \tag{33}$$

This result implies that the failure probability under guided proposals vanishes exponentially faster than that of the unguided baseline as the budget $k$ increases.

### D.4. Further Discussion

While the hitting-time analysis focuses on the number of evaluations ($N$), practical efficiency depends on the total wall-clock time. Let $C_{\text{infer}}$ be the inference cost to generate one proposal and $C_{\text{eval}}$ be the cost to verify it (including BFGS optimization). Since $N$ follows a geometric distribution, the expected number of trials is:

$$\mathbb{E}[N] = 1/p. \tag{34}$$

The total expected cost is:

$$\mathbb{E}[T] = \mathbb{E}[N] \times (C_{\text{infer}} + C_{\text{eval}}) = \frac{1}{p}(C_{\text{infer}} + C_{\text{eval}}). \tag{35}$$

In our framework, the guided inference cost $C_{\text{infer}}^\theta$ is marginally higher than the baseline $C_{\text{infer}}^0$ due to the overhead of guidance generation. However, in Symbolic Regression, verification is the dominant bottleneck, typically satisfying $C_{\text{eval}} \gg C_{\text{infer}}$. Since the expected number of evaluations is reduced by a factor of roughly $\frac{p_0}{p_0 + \gamma}$, the reduction in the expensive evaluation term significantly outweighs the marginal increase in inference overhead, leading to a net reduction in total computational cost.

## E. Experimental Settings and Implementation Details

### E.1. Benchmark Details

Traditional symbolic regression (SR) benchmarks, such as SRBench (La Cava et al., 2021) and SRSD (Matsubara et al., 2024), are primarily designed for evaluating classical or learning-based SR methods and are therefore suboptimal for rigorously assessing LLM-integrated methods. A key limitation is that many benchmark equations correspond to well-established canonical forms that are likely to appear in the pre-training data of LLMs. As a result, strong performance on these benchmarks may partially reflect memorization or pattern recall, rather than reflecting symbolic discovery abilities.

Based on this concern, we adopt LLM-SRBench (Shojaee et al., 2025b), a novel and challenging benchmark specifically designed for LLM-integrated SR methods. LLM-SRBench is meticulously constructed to mitigate trivial rote recitation while fully capitalizing on the scientific priors embedded within LLMs.

The benchmark consists of two complementary classes of problem sets: (i) **LLM-Transform**, which transforms existing benchmark equations into different mathematical representations, requiring models to reason beyond memorized forms. (ii) **LLM-Synth**, which combines known functional terms with synthetic, novel ones, further challenging models to extrapolate beyond familiar patterns. LLM-SRBench comprehensively covers four critical scientific domains: Physics, Chemistry, Biology, and Materials Science, ensuring broad applicability. In total, the benchmark comprises 239 distinct problem instances, providing a statistically robust foundation for evaluation. Detailed benchmark statistics are provided in Tab. 6 for reference.

*Table 6.* Statistics of the LLM-SRBench Benchmark Dataset

| | LSR-Transform | LSR-Synth | | | | Overall |
|---|---|---|---|---|---|---|
| | | Chemistry | Biology | Physics | Material | |
| **#Problems** | 111 | 36 | 24 | 44 | 25 | 240 |
| **Avg Datapoints (k)** | 94.7 | 5.0 | 5.0 | 5.0 | 5.0 | 46.5 |
| **Avg Variants** | 5.4 | 3.0 | 3.0 | 3.9 | 3.0 | 4.3 |

### E.2. Baseline Implementation Details

We compare against various representative LLM-based symbolic regression baselines. In detail, the baselines include:

**LLM-SR (Shojaee et al., 2025a)** employs LLMs as mutation operators within an evolutionary search process, aiming to utilize their scientific knowledge for hypothesis generation. In each iteration, the LLM is prompted with the problem context, sampled parent expressions, and their MSE to generate candidate offspring. The model generates expression skeletons in Python programs, and BFGS further optimizes the parameters. While effective in producing plausible symbolic forms, the exploration process is largely shaped by the LLM's generative priors, which may bias search trajectories and lead to premature convergence to local optima. Moreover, the method relies on a single, scalar metric for parent selection and refinement. Such ambiguous feedback provides limited insight into structural deficiencies of candidate equations, thereby constraining the optimization process to incremental trial-and-error rather than systematic improvement.

**LASR (Grayeli et al., 2024)** augments the PySR framework by integrating a library of abstract textual concepts, leveraging zero-shot LLM queries to discover and evolve hypotheses through a combination of evolutionary and LLM-guided steps. The search process is rendered inefficient due to the uncoordinated application of LLM-driven and random mutations. Concurrently, the selection of parent solutions based solely on fitness creates excessive selection pressure, which skews the search towards exploitation and stifles the diversity necessary for effective global exploration.

**Scientific Generative Agent (SGA) (Ma et al., 2024)** is structured as a bi-level optimization framework that integrates LLMs with simulations. In the outer loop, LLMs act as symbolic reasoners that generate scientific hypotheses based on observational feedback from simulations. These hypotheses are then evaluated through simulations in the inner loop. The inner loop, in turn, performs gradient-based optimization on continuous parameters via differentiable simulations. Although the method employs an "explore-and-exploit" strategy by modulating the LLMs' generation temperature, this mechanism alone may not sufficiently guide the search in complex scientific domains. Notably, SGA does not adopt an evolutionary approach; rather than iteratively refining prior solutions, it generates new candidate hypotheses from scratch in each cycle, without systematically leveraging experience from previously evaluated ones.

To ensure a rigorous and equitable evaluation, our experimental protocols strictly adhere to the standardized configurations established in both the original literature and LLM-SRBench. Specifically, to maintain a consistent computational budget, we impose a fixed discovery budget of 1,000 LLM-generated candidate samples per problem for LLM-based methods (LLM-SR, LASR, and SGA). For the hybrid LASR method, which synergizes LLM-informed operators with traditional symbolic search, we follow the established regime: besides the LLM-based samples, random non-LLM mutations are allocated to more than 453,000. The experiments use the vLLM framework (Kwon et al., 2023). All experiments are performed on four NVIDIA A800-SXM4-80GB GPUs. The comprehensive hyperparameter specifications and implementation details for all baselines are consolidated in Tab 7. The prompts are shown in Listing. 1.

### E.3. Deliberate Evolution Implementation Details

To facilitate reproducibility, we provide a comprehensive summary of the experimental configurations and hyperparameter specifications in Tab. 8. Furthermore, the complete implementation and source code will be made publicly available upon the publication of this work. We provide the prompt template in Listing. 2 and Listing. 3.

## F. Additionally Experimental Results

### F.1. Full Experiments

#### F.1.1. SYMBOLIC ACCURACY RESULTS

Following LLM-SRBench (Shojaee et al., 2025b), we additionally evaluate Symbolic Accuracy (SA) under the Qwen3-4B backbone. SA measures whether the recovered expression is symbolically equivalent to the ground-truth equation after simplification, and provides an assessment of symbolic correctness. We use the standard evaluation pipeline with GPT-4o-mini as the judge model.

As shown in Tab. 9, Deliberate Evolution achieves the highest overall SA among all methods, indicating that its gains are not limited to lower prediction error but also translate into more accurate recovery of the underlying symbolic structure. For example, on the Physics dataset, our method achieves 13.6% SA, compared with 9.1% for LLM-SR and 6.8% for LLMDirect.

*Table 7.* Implementation details and hyperparameters for LLM-based baselines.

| Method | Hyperparameter Configurations |
|---|---|
| Overall | LLM Temperature $T = 0.8$ 
 Max tokens = 8192 |
| LlmDirect | 5 equation program hypotheses sampled from LLM for initial prompt 
 Execution timeout threshold T = 30s per hypothesis 
 Constant refinement via SciPy-based BFGS optimizer |
| LLM-SR | Batch size $b = 4$ equation programs per prompt 
 Parallel evaluators $e = 4$ 
 Islands for evolving process $m = 10$ 
 In-context parent samples per prompt $k = 4$ 
 BFGS optimizer from Scipy for parameter optimization 
 Maximum 10 parameters for equation skeleton |
| LASR | Iterations $= 25$ 
 Cycles per iteration $= 550$ 
 Number of populations $= 10$ 
 Population size $= 33$ 
 Operators: $+, -, \times, \div, \wedge, \exp, \log, \sqrt{}, \sin, \cos, \tan, \cosh$ 
 LLM weights: llm_mutate = llm_crossover = llm_gen_random $= 0.005$ 
 Default remaining configuration of PySR |
| SGA | MSE-driven objective for agentic feedback loop 
 torch.optim.Adam for differential parameter optimization 
 PyTorch-based implementation of model and torch.nn.Module class |

*Table 8.* Implementation details and hyperparameters in Deliberate Evolution.

| Domain | Hyperparameter Configurations |
|---|---|
| Population | Number of islands $m = 4$ 
 Population capacity per island: 400 
 Island reset interval: 50 
 Maximum evolutionary budget: 100 generations 
 Offspring batch size per generation: 4 |
| Adaptive Selection | Mechanism: Fitness-proportional Boltzmann sampling 
 Boltzmann initial temperature: 0.5 
 Exponential cooling rate: 0.95 
 Annealing schedule: Exponential decay 
 Stagnation recovery Boltzmann temperature: 2.0 |
| LLM Generation | Generation temperature $T = 0.8$ 
 Max tokens = 8192 
 Parent expression per prompt $k = 1$ ($k = 2$ for $o_{\text{cross}}$) 
 Maximum refinement rounds per sample: 4 |
| Reflective Memory | Periodic update interval: 12 generation 
 Breakthrough improvement threshold: $\epsilon = 0.1$ 
 Elite exemplars in reflection context: 3 
 Failed samples in reflection context: 3 
 Stagnation detection: $k = 8$ 
 Distilled insights number: 3 |

```
[Instruction]
You are a helpful assistant tasked with discovering mathematical function structures
    for scientific systems.
Complete the 'equation' function below, considering the physical meaning and
    relationships of inputs.

import numpy as np
#Initialize parameters
MAX_NPARAMS = 10
params = [1.0]*MAX_NPARAMS

@evaluate.run
def evaluate(data: dict) -> float:
    """ Evaluate the equation on data observations."""

    # Load data observations
    inputs, outputs = data['inputs'], data['outputs']
    X = inputs
    # Optimize parameters based on data
    from scipy.optimize import minimize
    def loss(params):
        y_pred = equation(*X, params)
        return np.mean((y_pred - outputs) ** 2)
    loss_partial = lambda params: loss(params)
    result = minimize(loss_partial, [1.0]*MAX_NPARAMS, method='BFGS')
    # Return evaluation score
    optimized_params = result.x
    loss = result.fun
    if np.isnan(loss) or np.isinf(loss):
        return None
    else:
        return -loss

[Parent Samples]
def equation_v0($INPUT_VAR[0], ..., $INPUT_VAR[N], params):
    """ Mathematical function for {$OUTPUT_VAR_DESC}
    Args:
        $INPUT_VAR[0]: A numpy array representing observations of
    {$INPUT_VAR_DESC[0]}.
        ...
        $INPUT_VAR[N]: A numpy array representing observations of
    {$INPUT_VAR_DESC[N]}.
    params: Array of numeric constants or parameters to be optimized
    Return: A numpy array representing {$OUTPUT_VAR_DES} as the result of applying
    the mathematical function to the
    inputs.
    """
    # Parent example 1 logic as function body...

def equation_v1($INPUT_VAR[0], ..., $INPUT_VAR[N], params):
    # Equation example 2...

[Function to be completed]
@equation.evolve
def equation_v2(input1: np.ndarray, input2: np.ndarray, params: np.ndarray) ->
    np.ndarray:
    """Improved version of `equation_v1`."""
    # LLM need to complete from here
```

*Listing 1.* Inplementation of the Python-based prompt used in the LLM-SR baseline.

```
[Instruction]
You are an expert in symbolic regression and mathematical expression discovery. Your
    goal is to find a mathematically and physically sound relationship between input
    variables and the target output.

[Available Tools]
You have access to specific tools to assist your reasoning. The system will
    automatically execute these tools based on your current reasoning context.
1. **analyze_data**
   - **Purpose**: Analyze the statistical properties and distribution of the training
    data. Call this if you need to understand variable ranges or data patterns.
2. **analyze_residuals**...
3. **dimensional_check**...

[Tool Call Format]
To use a tool, simply enclose the function name within <tool> tags.

[Memory Insights]
- Insight 1: Consider using trigonometric functions for angle-dependent relationships
- Insight 2: ...

[Constraints]
- Allowed operators: +, -, *, /, **, sqrt, sin, cos, log, exp
- Constants: pi, e
- Use ONLY ALL the input variables.

[Output Format]
1. **Reasoning**: Always think inside `` first to determine your next step.
After completing your reasoning, choose ONLY ONE of the following actions (do not
    perform both):
(1) <tool>function_name</tool> (Use this if you need more info. Do NOT generate the
    result yourself.)
(2) <answer>\\boxed{expression}</answer> (Use this only when you are confident in the
    solution.)
```

*Listing 2.* System Prompt for Deliberate Evolution.

```
[Problem]
Find the mathematical function skeleton for p_d (the dipole moment), given data on Ef
    (the electric field), epsilon (the electric constant or permittivity of the
    medium), and theta (the angle between the dipole axis and the position vector).
    Express p_d as an explicit function of the other variables.

[Previous Attempts]
Attempt 1:
    Expression: 4.237757*Ef - 111.120419*epsilon + 72.618953*theta

[Operator]
Task: Mutate the parent expression by making structural changes. Generate a new
    mathematical expression that may achieve a lower MSE.
```

*Listing 3.* User Prompt for Deliberate Evolution.

*Table 9.* Symbolic Accuracy (SA, ↑, %) results under the Qwen3-4B-Instruct-2507 backbone. Bold and underlined values indicate the best and second-best performance, respectively.

| Method | LSR-Transform | Physics | Material | Chemistry | Biology |
|---|---|---|---|---|---|
| LLMDirect | 13.5 | 6.8 | 12.0 | 0.0 | 0.0 |
| LLM-SR | 9.9 | 9.1 | **24.0** | 2.8 | 16.7 |
| LASR | 8.1 | 4.5 | 4.0 | 0.0 | 0.0 |
| SGA | 6.3 | 2.3 | 0.0 | 0.0 | 4.2 |
| **Deliberate Evolution** | **18.0** | **13.6** | **24.0** | **5.6** | **20.8** |

*Table 10.* Detailed Experimental Results over three independent runs.

| Backbone | Method | Run 1 | Run 2 | Run 3 | Mean ± Std |
|---|---|---|---|---|---|
| **Qwen3-4B-Instruct-2507** | LLMDirect | 5.46e-2 | 7.24e-2 | 5.78e-2 | 6.16e-2 ± 6.00e-5 |
| | LLM-SR | 2.51e-3 | 2.05e-3 | 2.93e-3 | 2.50e-3 ± 1.31e-7 |
| | LASR | 6.04e-3 | 6.97e-3 | 6.13e-3 | 6.38e-3 ± 1.77e-7 |
| | SGA | 1.04e-1 | 1.32e-3 | 1.15e-1 | 1.17e-1 ± 1.33e-4 |
| | **Deliberate Evolution** | **4.37e-4** | **4.58e-4** | **3.87e-4** | **4.27e-4 ± 8.87e-10** |
| **Llama3.1-8B-Instruct** | LLMDirect | 9.95e-3 | 1.98e-2 | 7.24e-3 | 1.23e-2 ± 2.92e-5 |
| | LLM-SR | 3.00e-3 | 4.69e-3 | 3.99e-3 | 3.89e-3 ± 4.82e-7 |
| | LASR | 6.07e-3 | 7.05e-3 | 6.12e-3 | 6.41e-3 ± 2.04e-7 |
| | SGA | 1.55e-1 | 1.43e-1 | 1.68e-1 | 1.55e-1 ± 1.04e-4 |
| | **Deliberate Evolution** | **1.01e-3** | **8.71e-4** | **8.22e-4** | **9.00e-4 ± 6.03e-9** |

### F.1.2. DETAILED STATISTICS OF RUN-TO-RUN STABILITY

To evaluate the robustness and reproducibility of various methods, we conducted three independent experimental runs to account for stochastic variations in the generation and optimization processes. All hyperparameters remained consistent across these runs to isolate the impact of initialization noise. The results are shown in Sec. 4.3 in the main text. Here, we report the detailed performance, mean, and variance in Tab. 10.

As represented in the table, Deliberate Evolution demonstrates superior stability compared to all baselines. For instance, on the Qwen3-4B backbone, our method reduces the standard deviation by an order of magnitude compared to the LLMDirect baseline (from 9e3 to 3e5). Same trends are observed for Llama3.1-8B backbone. This indicates that Deliberate Evolution exhibits less sensitivity to random initialization, making it a more reliable choice for practical deployment. Also, our approach demonstrates higher average performance while maintaining better robustness to random.

### F.1.3. DETAILED STATISTICS OF OUT-OF-DISTRIBUTION EVALUATION

In this section, we provide the precise numerical breakdown of the Out-of-Distribution (OOD) evaluation visualized in Fig. 7. Tab. 11 details the performance metrics across four diverse scientific domains: physics, material science, chemistry, and biology. These statistics confirm that the integration of adaptive exploration and diagnostic tools allows Deliberate Evolution to capture robust, invariant structures that hold valid beyond the training distribution, effectively mitigating the overfitting observed in competing approaches.

### F.2. Qualitative Analysis

### F.2.1. EVOLUTION TRAJECTORY CASE STUDIES

We provide detailed case studies of Deliberate Evolution to illustrate, as shown in Fig. 9, 10, and 11.

*Table 11.* Detailed Statistics on OOD Data.

| Method | Physics | | Material Sci. | | Chemistry | | Biology | |
|---|---|---|---|---|---|---|---|---|
| | NMSE ($\downarrow$) | $Acc_{0.01}$ ($\uparrow$) | NMSE ($\downarrow$) | $Acc_{0.01}$ ($\uparrow$) | NMSE ($\downarrow$) | $Acc_{0.01}$ ($\uparrow$) | NMSE ($\downarrow$) | $Acc_{0.01}$ ($\uparrow$) |
| LLMDirect | 3.43e4 | 6.82 | 1.94e-1 | 72.00 | 5.98e6 | 0.00 | 1.18e2 | 4.17 |
| LLM-SR | 8.17e4 | 9.09 | 2.75e-1 | 72.00 | 6.57e2 | 11.11 | 2.24e1 | 12.50 |
| LASR | 1.08e4 | 9.09 | 3.25e-1 | 40.00 | 4.49e2 | 2.78 | 9.66e2 | 4.17 |
| SGA | 9.92e5 | 6.82 | 2.91e-1 | 40.00 | 1.55e3 | 2.78 | 1.83e5 | 4.17 |
| Deliberate Evolution | **1.97e3** | **15.91** | **4.85e-2** | **80.00** | **1.09e1** | **11.11** | **1.14e1** | **12.50** |

**[Problem]**
  - v: the object's velocity (output)
  - E_n: the total energy of an object
  - m: the relativistic mass of an object
  - c: the speed of light

**[Ground Truth Expression]**
  -c*sqrt(1 - c**4*m**2/E_n**2)

**[Reflective Memory]**
  1. The successful mutation from a linear to a square-root relativistic form demonstrates that introducing structured physical laws significantly improves fitness.
  2. Test perturbations involving dimensionless combinations like sqrt(m/c) or (E_n - c²m)/c to uncover hidden physical relationships that align with relativistic kinematics beyond the standard energy-mass relation.

**[Tool Feedback]**
  - (Residual Diagnostic)  Error EXPLODES as E_n approaches c²m (near light speed regime) (Heteroscedasticity corr: 0.95). Suggestion: Your current function (linear combination) cannot capture relativistic effects. The error pattern indicates a SQUARE ROOT transformation.

**[Prediction]**
  -1.0*c*sqrt(E_n**2 - c**4*m**2)/E_n

*Figure 9.* Case study (I.48.2_1_0).

**[Problem]**
- E_n: the nth energy level of the system (output)
- m: the mass of a particle in a quantum mechanical system
- h: the Planck constant
- d: the width or distance between the boundaries of the quantum well

**[Ground Truth Expression]**
h**2/(8*pi**2*d**2*m)

**[Reflective Memory]**
1. Explore inverse-power terms to break the stagnation in the search space and improve fit across varying parameter regimes.
2. Explore non-monotonic combinations to introduce controlled inflection points and improve fit across diverse mass and width regimes.

**[Tool Feedback]**
- (Residual Diagnostic) Error EXPLODES as value increases (Heteroscedasticity corr: 1.00). Suggestion: Your current function (likely Polynomial or simple Exp) grows too slowly or too smoothly. Try RATIONAL FUNCTIONS (Division) to model the singularity (e.g., 1/(input+c)).

**[Prediction]**
0.125*h**2/(pi**2*d**2*m)

*Figure 10.* Case study (III.15.14_1_0).

**[Problem]**
- v: the object's velocity (output)
- E_n: the total energy of an object
- m: the relativistic mass of an object
- c: the speed of light

**[Ground Truth Expression]**
-c*sqrt(1 - c**4*m**2/E_n**2)

**[Reflective Memory]**
1. The successful mutation from a linear to a square-root relativistic form demonstrates that introducing structured physical laws significantly improves fitness.
2. Test perturbations involving dimensionless combinations like sqrt(m/c) or (E_n - c²m)/c to uncover hidden physical relationships that align with relativistic kinematics beyond the standard energy-mass relation.

**[Tool Feedback]**
- (Residual Diagnostic) Error EXPLODES as E_n approaches c²m (near light speed regime) (Heteroscedasticity corr: 0.95). Suggestion: Your current function (linear combination) cannot capture relativistic effects. The error pattern indicates a SQUARE ROOT transformation.

**[Prediction]**
-1.0*c*sqrt(E_n**2 - c**4*m**2)/E_n

*Figure 11.* Case study (II.35.21_2_1).

