# OpenReview forum: "Deliberate Evolution:  Agentic Reasoning for Sample-Efficient Symbolic Regression with LLMs"
_ICML.cc/2026/Conference — ICML 2026 regular_

### Official Review · Reviewer_r85E · 2026-02-26

**Soundness:** 3
**Presentation:** 3
**Significance:** 3
**Originality:** 3
**Overall Recommendation:** 4
**Confidence:** 4

**Summary:**

This paper introduces an agentic framework designed to improve the sample efficiency of LLM-based Symbolic Regression (SR). Previous LLM-based SR methods often struggle with "strategic myopia" and "coarse-grained feedback," relying on simple scalar metrics that fail to provide directional guidance or localize structural errors. This paper proposes decoupling the candidate proposal process from explicit guidance through three core modules: Adaptive Operators for strategic directional control, Tool-Augmented Proposal for diagnostic feedback (utilizing data profilers and residual diagnostics), and Reflective Memory to leverage historical insights from past trajectories. Extensive testing on the LLM-SRBench benchmark demonstrates that Deliberate Evolution outperforms existing baselines, achieving up to 55% reduction in average NMSE while requiring only 40% of the typical sample budget.

**Compliance With Llm Reviewing Policy:**

Affirmed.

**Final Justification:**

My concerns have been adequately addressed.

**Key Questions For Authors:**

See above.

**Limitations:**

The authors have discussed the limitations. Additional limitations in Weaknesses section should also be addressed.

**Strengths And Weaknesses:**

*Strengths*

The paper is well-written and easy to follow. The figures are well-designed and facilitate understanding.

By integrating directional, diagnostic, and historical signals, the model moves beyond "blind" trial-and-error to a more "deliberate," goal-oriented discovery process. The designs are generally well-motivated.

The framework consistently outperforms prior baselines with higher sample efficiency. The empirical performance is strong.


*Weaknesses*

It's unclear how the method compares against the more recent baselines, such as AlphaEvolve (OpenEvolve) and ShinkaEvolve. Validating the method with stronger proprietary models would be beneficial.

The theoretical framework presented in Section 3.5 and Appendix F exhibits a loose coupling with the specific architectural innovations of the paper. While the 'hitting-time analysis' provides a mathematically sound justification for search efficiency, it functions as a generalized probabilistic model for any guided search process rather than a rigorous derivation of the 'Deliberate Evolution' components themselves.

The current implementation relies on a predefined set of evolutionary operators and primitive tools; expanding this set for more specialized scientific domains may require additional expert engineering.

While the authors provide a stability analysis in the appendix (Table 7), the primary experimental results presented in the main text (e.g., Table 2) appear to rely on a single experimental run (specifically matching 'Run 1' in the supplementary data). This presentation choice obscures the potential variance inherent in LLM-based stochastic optimization.

Related works also involving guided proposal and reflective memory are not adequately discussed, e.g. [1-3].

[1] Ye, H., Wang, J., Cao, Z., Berto, F., Hua, C., Kim, H., ... & Song, G. (2024). Reevo: Large language models as hyper-heuristics with reflective evolution. Advances in neural information processing systems, 37, 43571-43608.

[2] Agrawal, L. A., Tan, S., Soylu, D., Ziems, N., Khare, R., Opsahl-Ong, K., ... & Khattab, O. (2025). Gepa: Reflective prompt evolution can outperform reinforcement learning. arXiv preprint arXiv:2507.19457.

[3] Liu, Y., Li, J., Zhao, W. X., Lu, H., & Wen, J. R. (2025). Experience-Guided Reflective Co-Evolution of Prompts and Heuristics for Automatic Algorithm Design. arXiv preprint arXiv:2509.24509.

---

> ### Author Rebuttal · Authors · 2026-03-31
>
> Thanks for the thoughtful comments and suggestions. Please find our point-to-point response below.
>
> ### W1: Comparison with recent baselines and stronger models
> **Recent Baselines:** We extend experiments to include OpenEvolve and ShinkaEvolve on Physics and Material with Qwen3-4B, under the same settings as in the main paper. **Deliberate Evolution achieves the best performance on both datasets.**
>
> |Method|Phys NMSE|Phys Acc|Mat NMSE|Mat Acc|
> |-|-|-|-|-|
> | OpenEvolve|6.58e-3|9.09|1.59e-3|24.00|
> | ShinkaEvolve|9.32e-3|6.82|2.10e-3|32.00|
> | Deliberate Evolution|4.37e-4|15.91|1.47e-4|56.00|
>
> **Stronger models:** Under rebuttal time constraints, we test GPT-4o-mini on 3 representative problems, **Deliberate Evolution achieves the lowest NMSE on these problems.**
> ||I.44.4_2_0|PO21|MatSci22|
> |-|-|-|-|
> |LLMDirect|3.88e-4|9.03e-3|9.03e-5|
> |LLM-SR|1.03e-4|4.43e-6|7.21e-6|
> |LASR|7.86e-5|3.28e-5|1.31e-6|
> |Deliberate Evolution|7.06e-5|1.88e-7|1.04e-8|
>
> ### W2: Scope of theoretical analysis
> Thanks for the insightful comment. We would like to clarify that the theoretical framework (Sec.3.5 and App.F) is intended as a **principled abstraction** of guided symbolic search. Its purpose is to provide a conceptual efficiency rationale (**the "why"**), while Deliberate Evolution specifies the realization for SR (**the "how"**)
> - **Theory as a general principle.** The hitting-time analysis is intentionally stated at a general level: it formalizes the link between local proposal quality ($p_\theta$) and global expected search efficiency. It functions as a justification for the guided-search principle, rather than a mechanical, module-by-module derivation of our design.
> - **Connection to our architecture.** The coupling between the theory and our architecture can be established through the decomposition of success probability (Eq.21): our SR-specific components map naturally to the abstract probabilistic terms. In particular, diagnostic tools can increase $P(f\in F_{valid})$ by pruning physically/dimensionally invalid candidates, while adaptive operators and reflective memory bias the search toward high-potential functional forms.
>
> In the revised manuscript, we will clarify this scope more explicitly. Thanks again for the insightful comment!
>
> ### W3: Generalization and engineering effort
> We appreciate this concern. Deliberate Evolution is designed as a modular and domain-agnostic system to minimize the need for expert engineering when expanding to new scientific domains.
>
> - **General-purpose core:** The main guidance components are not tailored to a single domain. Specifically, operators, diagnostic tools (e.g., residual patterns and statistics), and memory are broadly useful across symbolic discovery tasks.
> - **Modular extensibility:** The framework is a modular design, enabling domain-specific transfer when needed. For example, domain-specific design can be incorporated by extending the tool library, while leaving the core optimization loop unchanged. This shifts the burden from hand-crafting search logic to providing lightweight plug-in tools.
> - **Practical evidence:** We further support the extensibility with real-world stress-strain experiments [4] without additional engineering, where our method achieves the best performance, indicating the practical transferability.
>
> |Method|ID-NMSE|OOD-NMSE|
> |:---|:---:|:---:|
> |LLMDirect|3.91e-1|1.20e0|
> |LLM-SR|1.44e-1| 6.34e-1|
> |LASR|2.52e-1|1.15e0|
> |SGA|3.95e0|1.84e0|
> |Deliberate Evolution| **1.11e-1**|**2.98e-1** |
>
> [4] Stress-strain data for aluminum 6061-T651 from 9 lots at 6 temperatures under uniaxial and plane strain tension
>
> ### W4: Stability across independent runs
> The current main-text table reports a fixed-seed run, which makes the comparison deterministic but does not explicitly expose run-to-run variance. To address this more directly, we additionally report mean±standard deviation over $N=3$ independent runs on Matsci with Qwen3-4B.
>
> The averaged results show that **Deliberate Evolution remains consistently superior on Material**, indicating that the gain is stable across independent runs. This complements the stability analysis on Physics reported in Table 7.
>
> In the revision, we will update the main-text presentation with mean/std across repeated runs.
>
> |Method|Run 1|Run 2|Run 3|Mean±Std|
> |-|-|-|-|-|
> |LLMDirect|1.42e-3|1.83e-3|2.59e-3|1.95e-3±5.94e-4|
> |LLM-SR|3.55e-3|4.01e-3|4.46e-3|4.01e-3±4.55e-4|
> |LASR|6.21e-4|5.97e-4|7.59e-4|6.59e-4±8.74e-5|
> |SGA|1.02e-2|9.86e-3|1.31e-2|1.11e-2±1.78e-3|
> |Deliberate Evolution|1.47e-4|1.58e-4|1.41e-4|1.49e-4±8.62e-6|
>
> ### W5: Related work discussion
> Thanks for pointing this out. We will include more detailed related work discussion in the revised manuscript, including guided proposal, reflective memory, and related optimization frameworks.
>
>
> We will incorporate the discussed results and clarifications into the revised manuscript. We thank the reviewer again for the constructive feedback.

---

> > ### Author Rebuttal · Reviewer_r85E · 2026-04-03
> >
> > My concerns have been adequately addressed.

---

> > > ### Author Response · Authors · 2026-04-03
> > >
> > > Hi Reviewer r85E,
> > >
> > > We are glad that our response has addressed your concerns. And we truly appreciate your kind acknowledgment, positive support, and thoughtful feedback. All relevant discussions and additional results will be carefully integrated into the final version of the manuscript.
> > >
> > > Best regards, Authors of #24824

---

### Official Review · Reviewer_qV4W · 2026-03-11

**Soundness:** 3
**Presentation:** 3
**Significance:** 3
**Originality:** 2
**Overall Recommendation:** 4
**Confidence:** 3

**Summary:**

The key problem considered by this manuscript is the sample inefficiency and lack of strategic guidance in existing LLM-based symbolic regression (SR) methods, which typically rely on coarse-grained scalar feedback (e.g., MSE) and treat optimization steps as isolated episodes. This study's central aspect pertains to the proposal of the "Deliberate Evolution" framework, which provides directional guidance through adaptive evolutionary operators, diagnostic signals via tool-augmented generation, and utilization of historical iterative experience through a reflective memory mechanism, thereby equipping LLM-based symbolic regression methods with explicit, structured guidance. Experiments on the LLM-SRBench benchmark demonstrate that the proposed method outperforms representative baseline methods (LLMDirect, LLM-SR, LASR, SGA) while utilizing only 40% of the sample budget.

**Compliance With Llm Reviewing Policy:**

Affirmed.

**Final Justification:**

I recommend the paper as borderline accept.

**Key Questions For Authors:**

1. Could the authors provide a wall-clock time comparison between Deliberate Evolution and the baselines? Specifically, how does the overhead of tool calls and memory management affect the total runtime per sample?
2. Can you provide a qualitative case study demonstrating the specific impact of the diagnostic tools? For instance, show an equation generated without tool feedback vs. one generated with residual diagnostic feedback.
3. Regarding the minor discrepancy in LLMDirect baseline metrics between Table 1 and Table 2, could you clarify if this stems from different random seeds or experimental settings?
4. How does the memory retrieval mechanism scale with the number of stored insights? Is there a risk of context window saturation affecting performance on longer tasks? Are there differential impacts between in-distribution and out-of-distribution experiments?
Is Assumption 7 (p_θ≥p_0+γ) empirically validated? Could you provide an estimate or lower-bound analysis of γ?

**Limitations:**

Yes

**Strengths And Weaknesses:**

## Soundness

**Strengths**
- The framework is technically sound, integrating **adaptive operators, diagnostic tools, and memory mechanisms** in a coherent design.
- Experiments cover **multiple domains and models**, providing relatively comprehensive evaluation.

**Weaknesses**
- **Experimental inconsistency:** Minor discrepancy in LLMDirect NMSE (Noise-Free) between Table 1 and Table 2 (3.55e-1 vs 3.54e-1) without explanation.
- **Limited model scope:** Only two medium-sized open-source models are evaluated (Llama3.1-8B, Qwen3-4B). Performance with stronger models (e.g., GPT-4 or 70B+) remains unknown.
- **Weak theoretical guarantees:** The key assumption (Eq.7, \(p_\theta \ge p_0 + \gamma\)) relies on heuristic justification without theoretical proof.
- **Incomplete experiments:** Robustness and ablation studies are conducted only on **Qwen3-4B** and limited datasets.

---

## Presentation

**Strengths**
- The paper is well-organized and written in **clear, professional English**.
- Figures and tables effectively support the claims.

**Weaknesses**
- **Memory retrieval strategy is unclear.**
  While the memory update mechanism is described, the **retrieval policy beyond FIFO** is not specified.

---

## Significance

**Strengths**
- The paper identifies three key limitations in LLM-based SR:
  - **Strategic myopia** → addressed with adaptive operators.
  - **Coarse-grained feedback** → addressed with tool-based diagnostics.
  - **Episodic isolation** → addressed with a memory mechanism.
- Provides a **practical engineering framework** with clear applicability to SR research.

**Weaknesses**
- **Limited impact scope:** Primarily restricted to the SR domain and dependent on the base LLM’s reasoning ability.
- **Missing wall-clock efficiency:** The paper reports **sample efficiency (40% budget)** but does not provide runtime comparison despite potential overhead from tools and memory.

---

## Originality

**Strengths**
- **Adaptive operators:** Well-formalized strategy selection with stagnation-aware control.
- **Memory mechanism:** Dual-trigger update strategy (periodic + breakthrough).
- **Tool invocation:** Domain-specific diagnostic tools (numerical, residual, dimensional) adapted for SR.

**Weaknesses**
- Primarily **system-level innovation** rather than theoretical innovation.
- Adaptive operator ideas are **well-established in evolutionary algorithms**, with no convergence guarantees provided.
- **Memory retrieval strategy is under-specified.**
- **Tool impact is not clearly demonstrated:** No concrete examples showing how diagnostic feedback changes generated equations.

---

> ### Author Rebuttal · Authors · 2026-03-31
>
> Thanks for the thoughtful comments and suggestions. Please find our point-by-point response below.
>
> ### W1 & Q3: Inconsistency between Table 1 & Table 2
> Thanks for pointing this out. This is a minor reporting inconsistency, with no changes to seeds or settings. The correct value is 3.55e-1; we will fix it in the revision.
> ### W2 & W4: Model scale and experimental coverage
> We add stronger-model, robustness, ablation results on more backbones/datasets.
>
> **Stronger models:** Due to time limits, we test GPT-4o-mini on 3 problems. **Deliberate Evolution achieves the lowest NMSE on these problems.**
> ||I.44.4_2_0|PO21|MatSci22|
> |-|-|-|-|
> |LLMDirect|3.88e-4|9.03e-3|9.03e-5|
> |LLM-SR|1.03e-4|4.43e-6|7.21e-6|
> |LASR|7.86e-5|3.28e-5|1.31e-6|
> |Deliberate Evolution|7.06e-5|1.88e-7|1.04e-8|
>
> **Robustness:** On Physics (Qwen/Llama), **Deliberate Evolution consistently achieves the lowest NMSE at all noise levels.**
> ||Noise-Free|$\sigma=0.01$|$\sigma=0.05$|
> |-|-|-|-|
> |Qwen3-4B||||||
> |LLMDirect|5.46e-2|8.50e-2|9.53e-2|
> |LLM-SR| 2.51e-3 | 3.63e-3 |5.41e-3|
> |LASR|6.04e-3|8.82e-3|1.83e-2|
> |Deliberate Evolution|4.37e-4|5.82e-4|7.02e-4|
> |Llama3-8B||||||
> |LLMDirect|9.95e-3|1.82e-2|2.16e-2|
> |LLM-SR|3.00e-3|6.95e-3|1.06e-2|
> |LASR| 6.07e-3|9.75e-3|1.53e-2|
> |Deliberate Evolution|1.01e-3|3.97e-3|6.99e-3|
>
> **Ablation:** On Physics and Matsci, **removing any component degrades performance.**
> |Component|Variant|Phys NMSE|Phys Acc| Mat NMSE|Mat Acc|
> |-|-|-|-|-|-|
> |**Qwen3-4B** Default||4.37e-4|15.91|1.47e-4|56.00|
> |Mem|w/o Mem|1.34e-3|9.52|1.84e-3|40.00|
> |Tool|w/o Tool|2.52e-2|4.55|3.60e-3|36.00|
> |Operator|Fixed Refine|8.69e-3|9.52|9.38e-3|32.00|
> ||Uniform|1.02e-2|6.82|1.03e-3|44.00|
> ||w/o Stagnation|7.69e-4|13.64|3.46e-4|40.00|
> |**Llama3-8B** Default||1.01e-3|11.36|2.89e-4|64.00|
> |Mem|w/o Mem|1.99e-3|6.82|5.84e-4|32.00|
> |Tool|w/o Tool|1.35e-2|2.27|3.60e-3|12.00|
> |Operator|Fixed Refine|4.71e-3|6.82|9.38e-3|24.00|
> ||Uniform|8.96e-3|2.27|1.03e-3|32.00|
> ||w/o Stagnation|3.71e-3|9.09|3.46e-3|16.00|
> ### W3 & Q4: Empirical support for Eq.7 & convergence
> **Experimental support.** Using Qwen3-4B, we compare guided/unguided proposals in 64 matched mutation states, with 8 offspring per state. Success is defined by NMSE$<\epsilon$ ($\epsilon\in\{0.1,0.01\}$). We compute paired gaps $\hat{\Delta}_i=\hat{p}_{\theta,i}-\hat{p}_{0,i}$, and apply a one-sided sign test for $H_0:P_r(\hat{\Delta}_i>0|\hat{\Delta}_i\ne0)=0.5$. The results strongly support Eq.7.
>
> **Estimate of $\gamma$.** We estimate $\hat{\gamma}=\frac{1}{64}\sum_i\hat{\Delta}_i$ and report one-sided 95% bootstrap lower bounds. Results below show positive $\gamma$ and 95% lower bounds.
>
> |$\epsilon$|Win rate|P-value|Mean $\gamma$|95% Lower bound|
> |-|-|-|-|-|
> |0.1|94.7%|2.1e-13|0.13|0.11|
> |0.01|97.3%|2.8e-10|0.07|0.06|
>
> **Convergence.** We do not claim a formal convergence guarantee, which is standard for evolutionary methods like AlphaEvolve; empirical convergence is observed (Fig.2).
> ### W5, W11 & Q4: Memory scalability, context overhead, and ID/OOD impact
> To clarify, memory is a **fixed-size evolving state,** not an ever-growing database.
> - **Scalability:** The memory pool has fixed capacity (e.g., 5 entries) and is updated periodically via FIFO (lines 233-244), so cost does not grow with task length.
> - **Context overhead:** Memory size is explicitly bounded and averages **136.3 tokens**, only 0.05% of Qwen's 262k-token context window.
> - **ID/OOD impact:** On Physics, **memory improves both ID/OOD results,** with larger OOD gains.
>
> ||ID NMSE|ID Acc|OOD NMSE|OOD Acc|
> |-|-|-|-|-|
> |w/o Mem|1.34e-3|9.52|4.93e4|6.92|
> |w/ Mem|4.37e-4|15.91|1.97e3|15.91|
> ### W7 & Q1: Time comparison
> We report wall-clock time and token usage per problem on Physics below.
> - **Deliberate Evolution is the fastest** despite tools/memory, with main cost from LLM generation.
> - **Tools/memory overhead is limited.** Tools are lightweight functions; extra cost comes from added LLM turns. Memory is called periodically with marginal overhead.
>
> ||Time(s)|Tokens(M)|
> |-|-|-|
> |LLMDirect|1919|1.34|
> |LLM-SR|5005|1.54|
> |LASR|4645|1.24|
> |SGA|7951|2.73|
> |Deliberate Evolution|1167|1.04|
> |Tool overhead|395|0.43|
> |Mem overhead|78|0.02|
> ### W12 & Q2: Qualitative case study of tools
> To isolate tool effects, we compare one mutation from the same evolutionary state w/ and w/o tools. We report symbolic skeletons (constants omitted).
>
> **PO10**
> - Gt: $\sin(t)+xt+x^3$
> - Tool feedback: Residuals correlate with $\sin(t)$ (corr:-0.67), suggesting a missing periodic term in $t$...
>
> ||Expression|MSE|
> |-|-|-|
> |Parent|$xt+x^3$|4.4e-1|
> |Ours|$\sin(t)+xt+x^3$|7.8e-12|
> |w/o tool|$xt+x^3+x^2$|1.2e-1|
>
> **CRK3**
> - Gt: $A^2+A\exp(t)+\cos(\log(A))$
> - Tool feedback: Residuals show a parabolic shape with A, suggesting missing $A^2$...
>
> ||Expression|MSE|
> |-|-|-|
> |Parent|$A+t$|6.3e-4|
> |Ours|$A^2+A\exp(t)$|2.5e-7|
> |w/o tool|$A^3+A+\sin(t)$|9.7e-4|
>
> These cases show that **diagnostic tools provide targeted guidance;** without them, edits become more trial-and-error.

---

> > ### Author Rebuttal · Reviewer_qV4W · 2026-04-03
> >
> > I would like to thank the authors for their responses. I will keep my positive score.

---

> > > ### Author Response · Authors · 2026-04-03
> > >
> > > Hi Reviewer qV4W,
> > >
> > > We are glad that our response has addressed your concerns. And we truly appreciate your kind acknowledgment, positive support, and thoughtful feedback. All relevant discussions and additional results will be carefully integrated into the final version of the manuscript.
> > >
> > > Best regards,
> > > Authors of #24824

---

### Official Review · Reviewer_o4Xq · 2026-03-11

**Soundness:** 3
**Presentation:** 3
**Significance:** 3
**Originality:** 3
**Overall Recommendation:** 4
**Confidence:** 4

**Summary:**

The paper proposes Deliberate Evolution, an LLM-based framework for symbolic regression that augments evolutionary search with structured guidance signals, including directional operators, diagnostic analysis, and trajectory memory. Experiments on symbolic regression benchmarks show improved sample efficiency and accuracy compared with prior LLM-driven approaches.

**Compliance With Llm Reviewing Policy:**

Affirmed.

**Final Justification:**

After carefully considering the original manuscript and the authors’ comprehensive rebuttal, I have decided to upgrade my evaluation from Weak Reject to Weak Accept. The authors have put in a significant effort to address the technical gaps identified in the initial review, particularly regarding empirical grounding and comparative analysis.

The authors provided the requested ablation studies, which clarify the individual contributions of directional operators, diagnostics, and trajectory memory. Furthermore, the inclusion of comparisons against classical GP-based symbolic regression systems and results on real-world datasets strengthens the paper’s claims.

The originality and novelty remain the primary point of contention. However, I recognize that the integration of these specific signals for the symbolic regression domain is well-executed and provides a practical "Industry Brain" style of engineering contribution.

The authors have pledged to include more implementation details and release the code, which addresses my concerns regarding reproducibility.

In conclusion, the paper now presents a more complete and rigorous study. The strengths in empirical performance and clarity now outweigh the limitations in conceptual novelty, making it suitable for publication.

**Key Questions For Authors:**

1.	The framework includes several guidance signals (directional operators, diagnostic analysis, and trajectory memory). Can the authors provide a detailed ablation study isolating the contribution of each component?
2.	The evaluation primarily compares against LLM-based symbolic regression approaches. How does the proposed method perform against strong classical symbolic regression systems (e.g., modern GP-based approaches)?
3.	While the paper emphasizes improved sample efficiency, the framework introduces additional analysis and guidance steps. What is the total computational cost compared with baseline methods (e.g., wall-clock time or total LLM calls)?
4.	The experiments focus on standard symbolic regression benchmarks. Have the authors evaluated the approach on more complex or real-world symbolic discovery tasks?
5.	Some details of the guidance mechanisms and optimization loop are not fully specified. Can the authors clarify the implementation details and whether code will be released?

**Limitations:**

yes

**Strengths And Weaknesses:**

Strengths

Soundness.
Experiments on standard symbolic regression benchmarks suggest improved sample efficiency over existing LLM-based approaches.

Presentation.
The paper is clearly structured and presents the framework and experimental results in a generally easy-to-follow manner.

Significance.
Improving search efficiency in LLM-driven symbolic regression is a relevant problem with potential implications for symbolic discovery systems.

Originality.
The work integrates multiple guidance signals into an LLM-driven evolutionary framework for symbolic regression.

Weaknesses

1.	The core idea, augmenting LLM generation with additional feedback signals, is conceptually similar to existing agentic optimization and iterative refinement frameworks. The novelty mainly lies in combining known mechanisms rather than introducing fundamentally new techniques.

2.	The framework introduces several guidance signals (directional operators, diagnostics, and trajectory memory), but the paper does not clearly isolate their individual contributions. More detailed ablation studies are necessary to understand which components actually drive the reported improvements.

3.	Experiments focus primarily on symbolic regression benchmarks. It remains unclear whether the approach generalizes to more complex symbolic discovery settings or real scientific problems.

4.	The evaluation largely compares against other LLM-based methods, while stronger classical symbolic regression approaches (e.g., modern GP-based systems) are not extensively analyzed.

5.	Although the paper reports improved sample efficiency, the evaluation does not fully account for the computational cost introduced by additional analysis and guidance steps.

6.	Some implementation details of the guidance mechanisms and optimization loop are not fully specified, which may hinder reproducibility.

---

> ### Author Rebuttal · Authors · 2026-03-31
>
> Thanks for the thoughtful comments. Please find our point-by-point response below.
> ### W1: Novelty from compositional design
> We would like to clarify that our contribution lies in a **SR-specific architecture shift**: from black-box proposal to deliberate optimization.
> - **SR-specific bottleneck:** Prior LLM-based SR methods rely on (parent, MSE). However, scalar MSE is inherently ambiguous in SR: it cannot distinguish structural flaws from numerical errors, forcing baselines into myopic trial-and-error.
> - **Unified guidance architecture:** We reformulate the process by explicitly decoupling proposal from guidance, and by introducing complementary signals tailored to SR:
>
> |Feature|SR Baselines|Ours|
> |-|-|-|
> |Error signal|Scalar MSE|Diagnostic feedback|
> |Search control|Implicit LLM prior|Adaptive operators|
> |History|Isolated|Reflective memory|
> - **Empirical support.** Experiments support this design: in addition to better sample efficiency and final performance, ablations show that each component contributes.
> ### W2 & Q1: Ablation of each component
> To clarify, we have conducted ablations on each component and summarized the results in Table 3: tool/memory are removed directly to isolate diagnostic/historical guidance; directional guidance is ablated by removing operator diversity (Fixed Refine), removing adaptive selection (Uniform), and removing stagnation-aware control (w/o Stagnation).
>
> We add additional ablations on Material with Qwen3-4B. As shown below, **removing or simplifying any component consistently degrades performance.** On Physics, NMSE increases from 4.37e-4 to 2.52e-2 without tools. Similar trends hold on Material.
> |Component|Variant|Phys NMSE|Phys Acc|Mat NMSE|Mat Acc|
> |-|-|-|-|-|-|
> |Default||4.37e-4|15.91|1.47e-4|56.00|
> |Mem|w/o Mem|1.34e-3|9.52|1.84e-3|40.00|
> |Tool|w/o Tool|2.52e-2|4.55|3.60e-3|36.00|
> |Operator|Fixed Refine|8.69e-3|9.52|9.38e-3|32.00|
> ||Uniform|1.02e-2|6.82|1.03e-3|44.00|
> ||w/o Stagnation|7.69e-4|13.64|3.46e-4|40.00|
> ### W3 & Q2: Comparison with classical SR
> We add strong classical SR methods: GPlearn, PySR [1], DSR [2], and uDSR [3]. Following prior work, we use standard implementations and allow up to 2M iterations (vs. only 400 for ours). Due to rebuttal time limits, we report Physics and Material and will add the full comparison in revision.
>
> **Deliberate Evolution outperforms baselines despite a far smaller search budget.** On Physics, our method w/ Qwen3-4B achieves 4.37e-4 NMSE compared to 2.50e-3 for PySR. Similar trends hold on Material and across both backbones.
> ||Phys NMSE|Phys Acc|Mat NMSE|Mat Acc|
> |-|-|-|-|-|
> |GPlearn|5.86e-3|4.55|4.67e-3|0.00|
> |PySR|2.50e-3|11.36| 8.67e-3|4.00|
> |DSR|2.99e-1|2.27|1.05e1|0.00|
> |uDSR|2.07e-2|4.55|2.13e1|0.00|
> |Deliberate Evolution w/ Llama3-8B|1.01e-3|11.36|2.89e-4|**64.00**|
> |Deliberate Evolution w/ Qwen3-4B|**4.37e-4**|**15.91**|**1.47e-4**|56.00|
>
> [1] Interpretable Machine Learning for Science with PySR and SymbolicRegression.jl
> [2] Deep symbolic regression: Recovering mathematical expressions from data via risk-seeking policy gradients
> [3] A Unified Framework for Deep Symbolic Regression
> ### W4 & Q3: Cost comparison
> We compare computational cost on Physics as shown below. **Among LLM-based methods, Deliberate Evolution is the fastest with minimal tokens,** indicating efficient search under deliberate guidance. For non-LLM methods, PySR is significantly faster, likely due to its Julia implementation.
> ||Time(s)|Calls|Tokens(M)|
> |-|-|-|-|
> |LLMDirect|1919|1k|1.34|
> |LLM-SR|5005|1k|1.54|
> |LASR|4645|1k|1.24|
> |SGA|7951|1k|2.73|
> |Deliberate Evolution|1167|723|1.04|
>
> ||Time(s)|
> |-|-|
> |GPlearn|6943|
> |PySR|484|
> |DSR|8862|
> |uDSR|9025|
> ### W5 & Q4: Real-world tasks
> We evaluate on the Stress-Strain dataset [4], a real-world symbolic discovery task on aluminum 6061-T651 measurements. The setup follows the main paper. We report in-domain (ID) and out-of-domain (OOD) NMSE, omitting Acc since noisy real-world data makes exact symbolic matching less informative.
>
> As shown in the table below, **Deliberate Evolution achieves the best ID/OOD NMSE:** e.g., it achieves 1.11e-1 ID-NMSE compared with 1.44e-1 for LLM-SR; it achieves 2.98e-1 on OOD data, lower than baselines.
>
> ||ID-NMSE|OOD-NMSE|
> |-|-|-|
> |LLMDirect|3.91e-1|1.20e0|
> |LLM-SR|1.44e-1|6.34e-1|
> |LASR|2.52e-1|1.15e0|
> |SGA|3.95e0|1.84e0|
> |GPlearn|3.57e-1|7.45e-1|
> |PySR|1.71e-1|5.68e-1|
> |DSR|1.91e0|1.61e2|
> |uDSR|5.78e-1|1.51e1|
> |Deliberate Evolution|1.11e-1|2.98e-1|
>
> [4] Stress-strain data for aluminum 6061-T651 from 9 lots at 6 temperatures under uniaxial and plane strain tension
> ### W6 & Q5: Reproducibility
> To clarify the loop in Alg.1: each iteration samples a parent, adaptively selects a directional operator (Sec.3.2), generates and evaluates a child with tools (Sec.3.3), updates the population, and periodically updates memory (Sec.3.4). We provide anonymous code [5], and will expand implementation details in the revision.
>
> [5] https://anonymous.4open.science/r/Deliberate-Evolution

---

> > ### Author Rebuttal · Reviewer_o4Xq · 2026-04-01
> >
> > Thank you for the clear and thorough rebuttal. The additional ablations, classical SR comparisons, and real-world experiments improve the paper’s completeness and address most of my questions. However, my main concern about limited conceptual novelty remains, as the method appears to be a well-engineered integration rather than a fundamentally new approach. Therefore, I maintain my original score.

---

> > > ### Author Response · Authors · 2026-04-02
> > >
> > > We sincerely thank Reviewer o4Xq for the thoughtful comments and for recognizing our solid empirical study, clear presentation, and practical significance.
> > >
> > > In response to the question about conceptual novelty, we would like to clarify that **Deliberate Evolution** should be viewed as a **principled reformulation of LLM-based symbolic regression (SR) through grounded guidance.**
> > >
> > > While some individual components of methods have precedents in prior work, we believe our contributions go beyond adopting them in isolation: we **identify** a core SR-specific bottleneck, **reformulate** the search paradigm, **instantiate** it through a unified SR-specific guidance architecture, and **validate** its effectiveness through SOTA performance in SR.
> > >
> > > We respectfully elaborate on this point in more detail below.
> > >
> > > **(1) Diagnosis: the scalar-feedback bottleneck.** Prior LLM-based SR methods typically follow a *generate-evaluate-regenerate* loop [1, 2], where evaluation is reduced to a scalar signal (e.g., MSE). This formulation implicitly assumes that:
> > >   - (a) the evaluation signal already contains sufficient information for effective revision;
> > >   - (b) the LLM can reliably convert that signal into effective self-refinement.
> > >
> > > **However, in our view, both assumptions are limited in SR.** (a) MSE is inherently ambiguous: similar errors may arise from distinct causes, and therefore require different edits. (b) Prior studies [3, 4] also suggest that LLMs do not reliably self-refine under insufficiently grounded external feedback. As a result, relying on scalar evaluation alone is a key limitation of current LLM-based SR.
> > >
> > > **(2) Reformulation: from scalar feedback to guided generation.** The limitation above suggests that, in SR, candidate generation should be guided by explicit, externally grounded signals rather than by scalar evaluation alone.
> > >
> > > Accordingly, the search process is reformulated as *generate-diagnose-guide-regenerate.* This shift is not merely procedural; it reflects a different view of SR, in which effective optimization requires not only candidate generation, but also grounded guidance for targeted improvement.
> > >
> > > **(3) Instantiation: a unified guidance architecture for SR.** We instantiate this reformulation through three complementary forms of guidance:
> > >  - (i) **directional guidance**, which specifies **how** the next edit should proceed;
> > >  - (ii) **diagnostic guidance**, which explains **why** the current candidate fails;
> > >  - (iii) **historical guidance**, which preserves **what** useful structure or search experience should persist across iterations.
> > >
> > > Together, these components form a unified guidance architecture that provides a richer search state for SR: the past is preserved, the present is diagnosed, and the future edit is directed.
> > >
> > > **(4) Validation: the empirical evidence.** We evaluate Deliberate Evolution against both classical and LLM-based baselines, and it achieves SOTA performance in SR. Further analyses of sample efficiency, out-of-distribution generalization, run-to-run stability, robustness to observation noise, and ablations consistently support its effectiveness, robustness, and reliability.
> > >
> > > **Collectively, we hope this clarifies that our work should be characterized as a principled reformulation of LLM-based SR through deliberate guidance, rather than as an engineering combination of existing mechanisms.**
> > >
> > > References:
> > > [1] LLM-SR: Scientific Equation Discovery via Programming with Large Language Models
> > > [2] LLM and Simulation as Bilevel Optimizers: A New Paradigm to Advance Physical Scientific Discovery
> > > [3] Large Language Models Cannot Self-Correct Reasoning Yet
> > > [4] Pride and Prejudice: LLM Amplifies Self-Bias in Self-Refinement

---

### Official Review · Reviewer_uiGL · 2026-03-13

**Soundness:** 3
**Presentation:** 4
**Significance:** 2
**Originality:** 3
**Overall Recommendation:** 4
**Confidence:** 4

**Summary:**

The paper proposes Deliberate Evolution, a framework to improve the sample efficiency of LLM-based symbolic regression. Instead of relying solely on scalar feedback, it introduces structured guidance through adaptive operators, diagnostic tools, and a reflective memory module. Experiments on LLM-SRBench show improved accuracy and robustness while using fewer samples than state-of-the-art baselines

**Compliance With Llm Reviewing Policy:**

Affirmed.

**Final Justification:**

After considering the authors’ rebuttal, I find that the majority of my initial concerns have been satisfactorily addressed. The experimental evaluation demonstrates improvements over the chosen baselines. That said, the core contribution still feels somewhat incremental in relation to existing literature. So, I remain my original score as weak accept (4)

**Key Questions For Authors:**

check weaknesses above

**Limitations:**

yes the authors have provided a comprehensive discussion on the limitations of the proposed method and potential future research directions.

**Strengths And Weaknesses:**

## Strengths

- In general, the paper is well written and the results are clearly presented.
- The results appear competitive with state-of-the-art baselines.
- The evaluation is comprehensive and includes the standard benchmark in this field, LLM-SRBench, which contains more than 200 equation discovery problems.


## Weaknesses

**Major Concerns:**
- The description of the methodology is vague in several places. Currently it’s somewhat difficult to reproduce the method and its steps from the writing.
- It seems that the anonymous version of the code also has not been shared with the reviewers in the submission. I would encourage authors to do so as reproducibility is an important factor.
- I would suggest authors to also
    - include qualitative comparison of several discovered equations across methods, and
    - add symbolic accuracy (SA as in LLM-SRBench paper) to the evaluation metrics for comparison. The symbolic equivalence and proximity of the equation candidates are important for the purpose of this task.


**Minor Concerns:**
- The naming of the ablation variants in Table 3 is confusing and makes it difficult to clearly differentiate the variants.

---

> ### Author Rebuttal · Authors · 2026-03-31
>
> Thanks for the thoughtful comments and suggestions. Please find our point-by-point response below.
>
> ### Q1: Reproducibility
> > The description of the methodology is vague in several places. An anonymous version of the code is encouraged to be released.
>
> We have now provided an anonymous code release [1], which includes environment setup, full pipeline, and default hyperparameters. We will also revise the manuscript to improve reproducibility by clarifying the step-by-step procedure, key implementation details, and core design choices.
>
> [1] https://anonymous.4open.science/r/Deliberate-Evolution
>
> ### Q2: Qualitative comparison of discovered equations
>
> Thank you for this helpful suggestion. We provide two representative qualitative comparisons across methods. For clarity, we omit fitted constants and compare the symbolic skeletons (i.e., variable-operator structure).
>
> | Task (Domain) | Method | Expression |
> |:--- |:--- |:--- |
> | **PO3** | **Ground-truth** | $\sin(t) + \sin(v) + v$ |
> | (Physics) | LLMDirect | $\int v(t) dt + v + \cos(t) + \sin(t)$ |
> | | LLM-SR | $\sin(t) + v + v^2 + v^3$ |
> | | LASR | $v + t$ |
> | | SGA | $v + v^2$ |
> | | **Ours** | $\sin(t) + \sin(v) + v$ |
> | | | |
> | **CRK4** | **Ground-truth** | $A^2 + A\log(t+1)$ |
> | (Chemistry) | LLMDirect | $t + A + 1$ |
> | | LLM-SR | $A + \frac{A^n}{1 + (A/K)^n} + A^2 + tA + 1$ |
> | | LASR | $A^2 \sin\!\left(\left(e^t + \frac{1}{A}\right)^{-A}\right)$ |
> | | SGA | $A^2 + At + 1$ |
> | | **Ours** | $A^2 + A\log(t+1) + Ae^{-t}$ |
>
> These examples show that:
> - **Deliberate Evolution more accurately recovers the underlying symbolic structure.** On **PO3**, it exactly recovers the ground-truth skeleton, while the baselines miss the key operator $\sin(v)$ or replace it with polynomial or linear surrogates. On **CRK4**, our method recovers the two dominant components, $A^2$ and $A\log(t+1)$, whereas the baselines tend to oversimplify the relation or introduce mismatched terms.
>
> - **Baselines often rely on surrogate fitting rather than mechanism recovery.** They match the observations through simpler or compensatory terms, without recovering the true governing form. This distinction is important in symbolic regression, where recovering a concise and faithful symbolic skeleton is central to the task.
>
> These examples are further supported in **Q3** by symbolic accuracy as a dataset-level metric. Together, they show that Deliberate Evolution more faithfully recovers the underlying symbolic skeleton. This pattern is also consistent with the role of deliberate guidance in steering a deliberate exploration process.
>
>
> ### Q3: Adding symbolic accuracy (SA)
> Following LLM-SRBench, we add symbolic accuracy (SA) to directly evaluate symbolic correctness. We use the standard evaluation pipeline with GPT-4o-mini as the judge model, which compares the symbolic form of the prediction against the ground-truth equation for mathematical equivalence.
>
> The following table represents SA (%) results on five subsets, where **bold** denotes the best result and *italics* the second-best. As shown in the table, **Deliberate Evolution achieves the best or tied-best SA on all five subsets.** For example, on the Physics dataset, our method achieves 13.6% SA, compared with 9.1% for LLM-SR and 6.8% for LLMDirect.
>
> This quantitative result is consistent with the qualitative comparisons in **Q2**, showing that Deliberate Evolution more accurately recovers the underlying symbolic skeleton.
>
> | Method | LSR-Transform | Physics | Material | Chemistry | Biology |
> |--------|--------------------|---------|----------|-----------|---------|
> | LLMDirect | *13.5* | 6.8 | 12.0 | 0.0 | 0.0 |
> | LLM-SR | 9.9 | *9.1* | **24.0** | *2.8* | *16.7* |
> | LASR | 8.1 | 4.5 | 4.0 |0.0  |  0.0|
> | SGA | 6.3 | 2.3 | 0.0 | 0.0 | 4.2 |
> | Deliberate Evolution | **18.0** | **13.6** | **24.0** | **5.6** | **20.8** |
>
> ### Q4: Unclear variant naming in Table 3
> > (Minor) The naming of the ablation variants in Table 3 is confusing and makes it difficult to clearly differentiate the variants.
>
> Thanks for pointing this out. Table 3 reports ablations of the main components in Deliberate Evolution. Specifically, tools and memory are removed directly to isolate **diagnostic and historical** guidance. For **directional** guidance, we ablate it along three axes: (1) removing operator diversity (Fixed Refine), (2) removing adaptive operator selection (Uniform), and (3) removing stagnation-aware control (w/o Stagnation). In the revised manuscript, we will revise Table 3 to make each ablation setting explicit and easier to understand.

---

> > ### Author Rebuttal · Reviewer_uiGL · 2026-04-02
> >
> > I appreciate the authors’ response; most of my concerns are addressed in rebuttal. The experiments are solid and demonstrate effectiveness over baselines. However, the overall contribution appears somewhat incremental relative to prior work, so I will maintain my weak accept score.

---

> > > ### Author Response · Authors · 2026-04-03
> > >
> > > Hi Reviewer uiGL,
> > >
> > > We are glad that our response has addressed your concerns. And we truly appreciate your kind acknowledgment, positive support, and thoughtful feedback. All relevant discussions and additional results will be carefully integrated into the final version of the manuscript.
> > >
> > > Best regards,
> > > Authors of #24824

---

### Decision · Program_Chairs · 2026-04-30

**Decision:**

Accept (regular)

**Comment:**

This paper proposes Deliberate Evolution, an agentic framework for LLM-based symbolic regression that replaces coarse scalar feedback with three forms of structured guidance: adaptive evolutionary operators for directional control, diagnostic tools for localized error analysis, and reflective memory for historical insight. The framework achieves consistent improvements over LLM-SR, LASR, and SGA baselines on LLM-SRBench while using only 40% of the sample budget.
All four reviewers recommend weak accept. They consistently praised the clear writing, comprehensive evaluation across multiple scientific domains and backbone models, and the strong empirical results, including OOD generalization, noise robustness, and run-to-run stability analyses. The rebuttal substantially strengthened the paper by adding symbolic accuracy metrics, comparisons against classical GP methods (PySR, DSR, uDSR), real-world stress-strain experiments, wall-clock time comparisons, and extended ablation studies across both backbones.
The primary concern shared across reviewers is that the conceptual novelty is limited, as the contribution is primarily a well-engineered integration of known mechanisms (adaptive operators, diagnostic tools, memory) rather than a fundamentally new technique. However, as Reviewer o4Xq noted in their updated justification, the integration is well-executed, and SR-specific, and the empirical gains are substantial enough to constitute a practical contribution. Given the unanimous positive consensus and the thorough experimental validation, the paper merits acceptance, though the authors should clearly contextualize the novelty relative to prior agentic optimization frameworks in the camera-ready version.